# An active allosteric mechanism in ASAP1-mediated Arf1 GTP hydrolysis redefines PH domain function

Olivier Soubias[1,2], Samuel L. Foley [3], Xiaoying Jian[2], Rebekah A. Jackson [1], Yue Zhang [1,7], Eric M. Rosenberg Jr [2], Benjamin J. Hu[4], Jess Li [1], Frank Heinrich[5,6], Margaret E. Johnson [3], Alexander J. Sodt [4], Paul A. Randazzo [2] ✉ & R. Andrew Byrd [1] ✉

GTPase-activating proteins are important regulators of small GTPases; among these, ASAP1 stimulates GTP hydrolysis on Arf1 and is implicated in cancer progression. ASAP1 contains a Pleckstrin Homology (PH) domain essential for maximum Arf·GTP hydrolysis. The prevailing view of PH domains is that they regulate proteins through passive mechanisms like membrane recruitment. In sharp contrast, we show that the PH domain of ASAP1 actively contributes to Arf1 GTP hydrolysis. By combining NMR, molecular dynamics simulations, kinetic assays, and mutational analysis, we find that the PH domain binds Arf·GTP at the membrane, to establish an active state primed for GTP hydrolysis. We identify key residues on the PH domain and Arf that drive this allosteric mechanism, which mathematical modeling shows contributes as much to GTPase activation as membrane recruitment. The finding that PH domains directly modulate small GTPases has broad implications for the Ras and Rho oncoprotein families.

Adenosine diphosphate–ribosylation factors (Arfs) are a family of GTPases that control membrane traffic, cytoskeletal dynamics and lipid signaling. The function of Arfs, which have no detectable intrinsic GTPase activity, is dictated by over 30 Arf guanosine triphosphatase–activating proteins (GAPs) that induce hydrolysis of Arf-bound GTP (Fig. 1A, top). The precise roles and the molecular basis for Arf interaction with the GAPs are still being discovered[1,2].

ASAP1 (ArfGAP with SH3 domain, Ankyrin repeat and PH domain 1) is a 130 kDa multidomain polypeptide composed of BAR (Bin/Amphiphysin/Rvs), PH (Pleckstrin Homology), Arf GAP (also referred to as Z), Ankyrin Repeat (also referred to as A), proline rich, and Src Homology 3 (SH3) domains that controls actin and cell adhesion

dynamics and is thought to contribute to invasion and metastasis in cancer[2] (Fig. 1A, bottom and SI1). A catalytic arginine in the Arf GAP domain is essential but not sufficient for maximum GTP hydrolysis. The isolated Arf GAP domain has low catalytic efficiency (-1 $M^{-1}sec^{-1}$). In contrast, a protein fragment containing the PH, Arf GAP and Ankyrin Repeat domains (PZA) acts as a robust GAP, with eight orders of magnitude greater efficiency (-10$^8$ $M^{-1}sec^{-1}$)[3,4]. PH domains are known to passively regulate protein function by (1) binding to phosphoinositides (PIPs) only (membrane recruitment) or to PIPs and proteins simultaneously (coincidence detection), thereby increasing local concentration and collision frequency, and (2) mediating autoinhibition by positioning protein elements to inhibit catalytic domains

[1]Center for Structural Biology, Center for Cancer Research, National Cancer Institute, National Institutes of Health, Frederick, MD, USA. [2]Laboratory of Cellular and Molecular Biology, Center for Cancer Research, National Cancer Institute, National Institutes of Health, Bethesda, MD, USA. [3]Department of Biophysics, The Johns Hopkins University, Baltimore, MD, USA. [4]Unit of Membrane Chemical Physics, Eunice Kennedy Shriver National Institute of Child Health and Human Development, National Institutes of Health, Bethesda, MD, USA. [5]Department of Physics Carnegie Mellon University, Pittsburgh, PA, USA. [6]NIST Center for Neutron Research, Gaithersburg, MD, USA. [7]Present address: Ring Therapeutics, Inc., Cambridge, MA, USA. ✉e-mail: randazzp@mail.nih.gov; byrdra@nih.gov

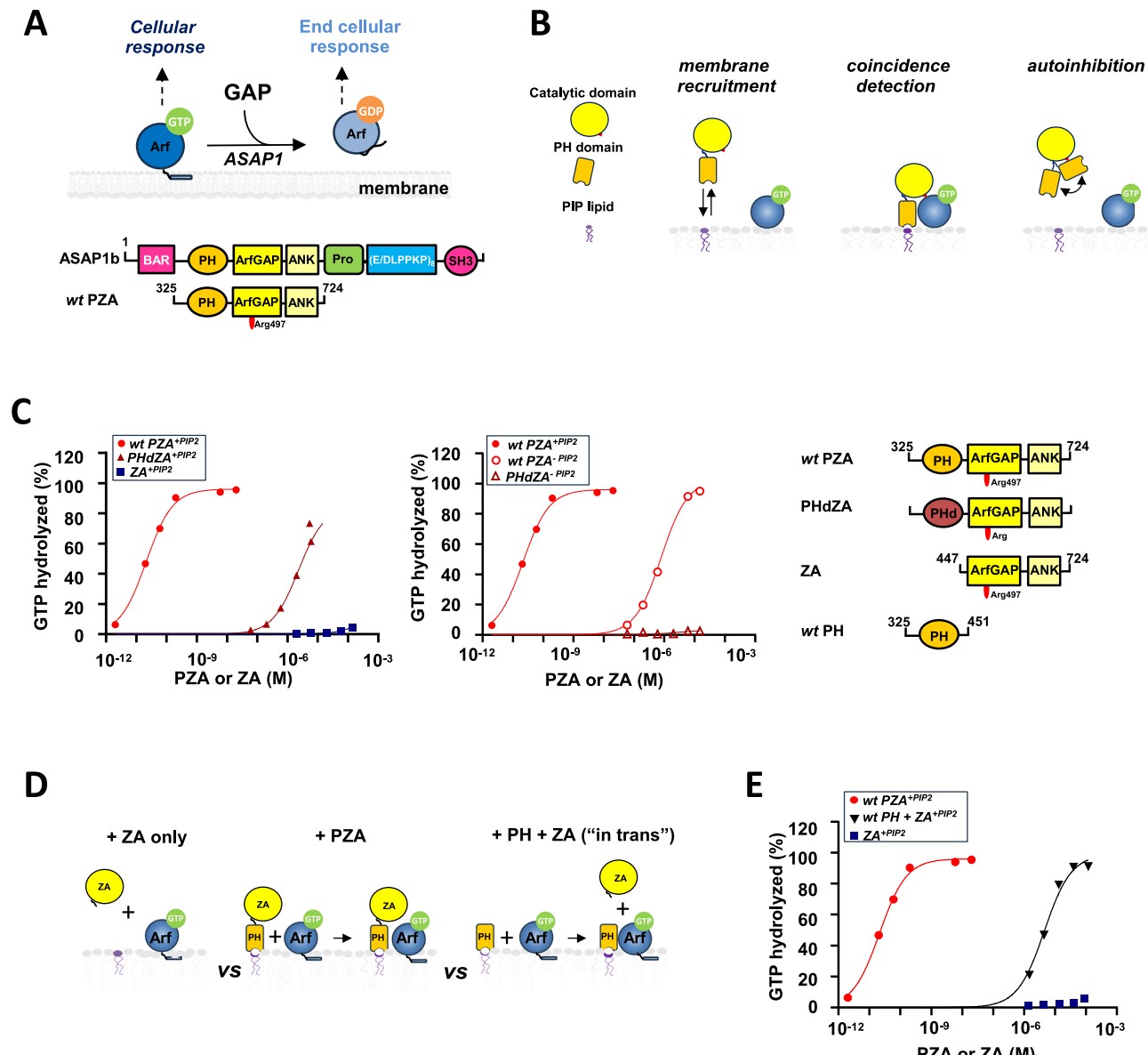

**Fig. 1 | Binding of ASAP1 PH to myrArf1 is key for maximal GTP hydrolysis.**
**A** Top. The ArfGAP ASAP1 terminates Arf signaling by catalyzing GTP hydrolysis. Bottom. Schematic of ASAP1 and ASAP1 catalytic unit. The domain structure of ASAP1 is shown in the schematic at top. BAR Bin/amphiphysin/RVS, PH pleckstrin homology, Arf Gap Arf GTPase-activating protein, ANK ankyrin repeat, Pro-Rich proline-rich, (E/DLPPKP)8 tandem repeats of E/DLPPKP, SH3 Src homology 3. Recombinant proteins used in the studies are shown below the schematic of full-length ASAP1. The acronyms for the proteins include "P" for the PH domain, "Z" for the Arf GAP domain, which is a zinc-binding motif and "A" for the ankyrin repeat. The catalytic arginine Arg497 is part of the Z domain. **B** Schematic of known passive mechanisms by which PH domains affect protein function. **C** (left). Comparison of GAP activity using Arf1 as substrate in the presence of PI(4,5)P$_2$ containing membranes. *wt* PZA (red circle), PHdZA (maroon ▲) or ZA (blue ■) was titrated into a GAP reaction containing 1 µM full-length Arf1 and LUVs at a total phospholipid concentration of 0.5 mM containing 5% mol PI(4,5)P$_2$. (middle). Comparison of GAP activity using Arf1 as substrate in the presence of membranes without PI(4,5)P$_2$ for *wt* PZA (red circle) and PHdZA (maroon open triangle). (right). Constructs design used in this study (see also Fig. SI1A). **D** Our approach to directly probe the role of the ASAP1 PH domain: comparing GAP activity using tandem constructs and isolated domains ("in trans" experiment). **E** The percentage of GTP bound to myr-Arf1 hydrolyzed in 3 min is plotted against the concentration of PZA (●), ZA (■) or ZA in the presence of 5 µM PH domain (▼). Source data are provided as a Source data file.

(autoinhibition) (Fig. 1B). We refer to these mechanisms as passive since they regulate the accessibility of another domain within the PH domain-containing protein to its target, but don't directly affect the target. While autoinhibition does not appear to be a regulatory mechanism in ASAP1, the molecular mechanism underlying the large PI(4,5)P$_2$-dependent increase in enzymatic activity remains unknown.

Here, we integrate Nuclear Magnetic Resonance (NMR), Molecular Dynamics (MD) simulations, and mathematical modeling of a large set of functional data to build a comprehensive structural-mechanistic model of the complex of Arf1 and the ASAP1 PH domain on a lipid

membrane surface, elucidate the underlying mechanisms by which the PH domain enhances GTP hydrolysis, and quantify the contribution of each mechanism to GAP activity. We find that the PH domain enhances catalytic activity by (i) remodeling the nucleotide binding site in Arf1, (ii) providing proximity and orientation to the catalytic residues in the Arf GAP domain by binding to its substrate Arf1·GTP, and (iii) restricting ASAP1 to a membrane surface to accelerate binding to the membrane associated substrate Arf1·GTP. The first two mechanisms primarily drive enzymatic activity. The active contribution to catalysis challenges the long-held assumption that PH domains act as passive

regulators. Our NMR biophysical and computational analyses identify key motifs on the PH domain (conserved in ArfGAPs) and Arf that mediate this allosteric effect. Taken together, our studies indicate that Arf undergoes protein-protein interaction-driven conformational changes that reorganize the nucleotide binding pocket of the small GTPase Arf1 and enhance GTP hydrolysis by orders of magnitude.

## Results

### ASAP1 PH contributes to catalysis beyond membrane recruitment

GAP activity of ASAP1 fragments were previously measured in our laboratories[5–8]. However, since these studies were done under different experimental conditions, we repeated the measurements in the presence of large unilamellar vesicles (LUVs) with or without $PI(4,5)P_2$ (5 mol%). The GAP activities of three constructs were compared: *wild-type* (*wt*) PZA, which includes the **P**H domain, **Z**inc-binding region, and **A**nkyrin repeats (residues 325–724 of ASAP1); PHdZA, a chimeric construct comprising the PLCδ1 PH domain in tandem with the ASAP1 ZA domain; and the $^{\Delta PH}$ZA domain (residues 447–724 of ASAP1, hereafter referred to as ZA), which lacks the PH domain and consequently does not interact with membranes. Both *wt* PZA and PHdZA exhibit efficient binding to $PI(4,5)P_2$ through their PH domains, with PHdZA displaying stronger membrane association than *wt* PZA. However, both constructs show negligible affinity for membranes lacking $PI(4,5)P_2$[5–8]. GAP activities depended on the protein constructs and on the presence of $PI(4,5)P_2$ (*wt* PZA$^{+PIP2}$ ≈ PHdZA$^{+PIP2}$ > *wt* PZA$^{-PIP2}$ > PHdZA$^{-PIP2}$ ≈ ZA$^{+PIP2 or -PIP2}$) indicating that although PH-dependent membrane recruitment contributes to GAP activity (*wt* PZA$^{+PIP2}$ > *wt* PZA$^{-PIP2}$), the cognate PH domain enhances GAP activity by other mechanisms, including binding to Arf (*wt* PZA$^{+PIP2}$ > PHdZA$^{+PIP2}$) (Fig. 1C and Table SI1).

To separate the effect of membrane recruitment from other activating mechanisms, we measured the rate of GTP hydrolysis "in trans" (Fig. 1D, E) using isolated PH and ZA domains. Strikingly, including the isolated ASAP1 PH domain in the reaction drastically increased activity over that observed with ZA alone, with ZA in the presence of the PH domain having a half-maximal effect at approximately $4 \times 10^{-6}$ M and inducing complete or near complete GTP hydrolysis at saturating concentrations (Fig. 1E). On the contrary, including the isolated PLCδ1 PH domain in the reaction did not enhance the rate of GTP hydrolysis over that observed with ZA alone (Fig. SI2A). We corroborated the results by tryptophan fluorescence and in the presence of nanodiscs (NDs)[9,10]. In the presence of ZA alone, GTP hydrolysis was not detected. Adding the ASAP1 PH domain to the reaction mixture with ZA triggered rapid GTP hydrolysis (Fig. SI2B, left). Similarly, the addition of the ASAP1 PH domain alone did not induce GTP hydrolysis, but subsequent addition of the isolated ZA domain initiated rapid GTP hydrolysis (Fig. SI2, middle). "In trans" activation experiments demonstrate that the ASAP1 PH domain's function extends beyond membrane recruitment; rather, they point to a mechanism where the ASAP1 PH domain induces conformational changes in Arf, increasing catalytic efficiency (Fig. 1E).

### Model of myrArf1:ASAP1 PH complex at the membrane surface

To gain insight into the allosteric mechanism, we sought to build a structural model of the complex between Arf1 and the PH domain of ASAP1 at the membrane surface. Given the challenges of X-ray crystallography and Cryo-EM for flexible membrane-associated systems of this size, we turned to NMR. In order to study a complex exceeding 100 kDa, we used methyl-labeled proteins, nanodiscs as a mimic for membrane bilayers, and Transverse Optimized Relaxation Spectroscopy (TROSY) techniques.

We first analyzed the spectral perturbations induced on a $^{1}$H-$^{13}$C methyl Heteronuclear Multiple Quantum Coherence (HMQC) spectrum of δ1-$^{13}$C$^{1}$H-labeled isoleucine (Ile), δ1/δ2 –$^{13}$C$^{1}$H-labeled Leucine (Leu) and γ1/γ2-$^{13}$C$^{1}$H-labeled Valine (Val) and otherwise perdeuterated myrArf1·GTPγS in the presence of an equimolar ratio of uniformly $^{2}$H (U-$^{2}$H) labeled ASAP1 PH at the surface of negatively charged NDs containing $PI(4,5)P_2$ (Figs. 2A, B and SI3A, B). In addition to the uniform broadening of myrArf1 resonances due to complex formation, specific chemical shift perturbations (CSPs) and selective resonance attenuation were seen on switch 1 (Val43, Ile49), switch 2 (Ile 74, Leu77) and the interswitch region (Val53) (Figs. 2C and SI3C). Those perturbations form a well-defined patch on one side of Arf1, suggesting that those residues are part of the interface with ASAP1 PH (Fig. 2D). Some smaller but significant changes were also seen on Leu 170 in C′ helix, which might result from indirect coupling effects between the two lobes of the G domain, and Leu 177 in the C′ helix and Val120 at the membrane facing tip of the β6 strand, which might result from interactions of those residues with lipids.

Chemical shift mapping data were complemented by comparing intensities of myrArf1 methyls in the presence of protonated ($I^{1H}$) or perdeuterated ($I^{2H}$) ASAP1 PH as in ref. 11 (Fig. 2E). A low $I^{1H}/I^{2H}$ indicates proximity between a methyl group and ASAP1 PH. The methyl resonance of interswitch residue Val53 broadened beyond detection when bound to protonated ASAP1 PH but was well resolved when bound to deuterated ASAP1 PH, indicating that Val53 is buried at the interface, in agreement with chemical shift mapping data. Smaller effects were observed for switch 1 (Ile42, Val43, Ile46 and Ile 49) and switch 2 residues (Ile74, Leu77), suggesting that those methyls are either only transiently part of the interface or not as deeply buried as Val53 (Fig. 2F).

To determine the surface of the PH domain in contact with Arf1, we monitored spectral perturbations on isotopically labeled *wt* PH when bound to an equimolar ratio of ND-associated myrArf1 in a $^{1}$H-$^{13}$C methyl HMQC spectrum. The *wt* PH domain was expressed as U-$^{2}$H,$^{15}$N and δ1-$^{13}$C$^{1}$H-labeled Ile, δ1 -$^{13}$C$^{1}$H-labeled Leu and γ1-$^{13}$C$^{1}$H-labeled Val, β -$^{13}$C$^{1}$H-labeled Alanine (Ala) and γ2-$^{13}$C$^{1}$H-labeled Threonine (Thr). The largest CSPs were observed on strands β5 (Thr387, Cys388, Gln389, Val390), β6 (Leu402) and β7 (Thr408) and the C-terminal α helix capping the β sandwich (Ile423, Leu434, Thr435 and Ala437) (Fig. 3A). Those perturbations form well-defined patches on one side of ASAP1 PH suggesting that strands β5-7 and the capping α helix are part of the interface with myrArf1 (Fig. 3B). Smaller CSPs on the other side of the PH domain i.e., at the tips of strands β2 and β3 (Val364, Ile368), suggests that a small fraction of *wt* PH might have a different orientation when interacting with myrArf1 (Fig. 3B).

Intermolecular paramagnetic relaxation enhancement[12,13] (PRE) effects induced on ASAP1 PH by spin-labeled myrArf1 variants were used to determine distance restraints between the interacting partners. To that end, two myrArf1 variants were obtained, in which Arf1 native cysteine (Cys159) was mutated to Ala and Lys38 (near switch 1), or Lys73 (near switch 2) were replaced with Cys to allow for spin-labeling. Their functional interaction with ASAP1 was largely unchanged compared to myrArf1 (Fig. SI4A). A nitroxide spin label (MTSL) was covalently linked to each of the two myrArf1 variants via a disulfide bond and used to measure PREs to methyl groups of ASAP1 PH. Ratios of intensities of ASAP1 PH resonances in the presence of each of the two spin-labeled variants in their paramagnetic ($I$, intensity with spin label) and diamagnetic ($I_0$, intensity with reduced spin label) states are shown in Figs. 3C and SI4B. Mapping the residues with $I/I_0 < 0.2$ (corresponding to distance <16 Å from the paramagnetic site location) onto the surface of ASAP1 PH (Fig. 3D) shows they localized within two patches (called hereafter β5/β7 patch and β2/β3 patch), on opposite sides of ASAP1 PH and overlapping with CSPs.

A model of the myrArf1:ASAP1 PH complex was built based on NMR experimental restraints by docking using the HADDOCK version 2.2. suite[14,15]. Because the two patches identified by PREs are separated by more than 20 Å, the observed PRE patterns cannot result from a single ensemble of similar orientations and also be consistent with CSP

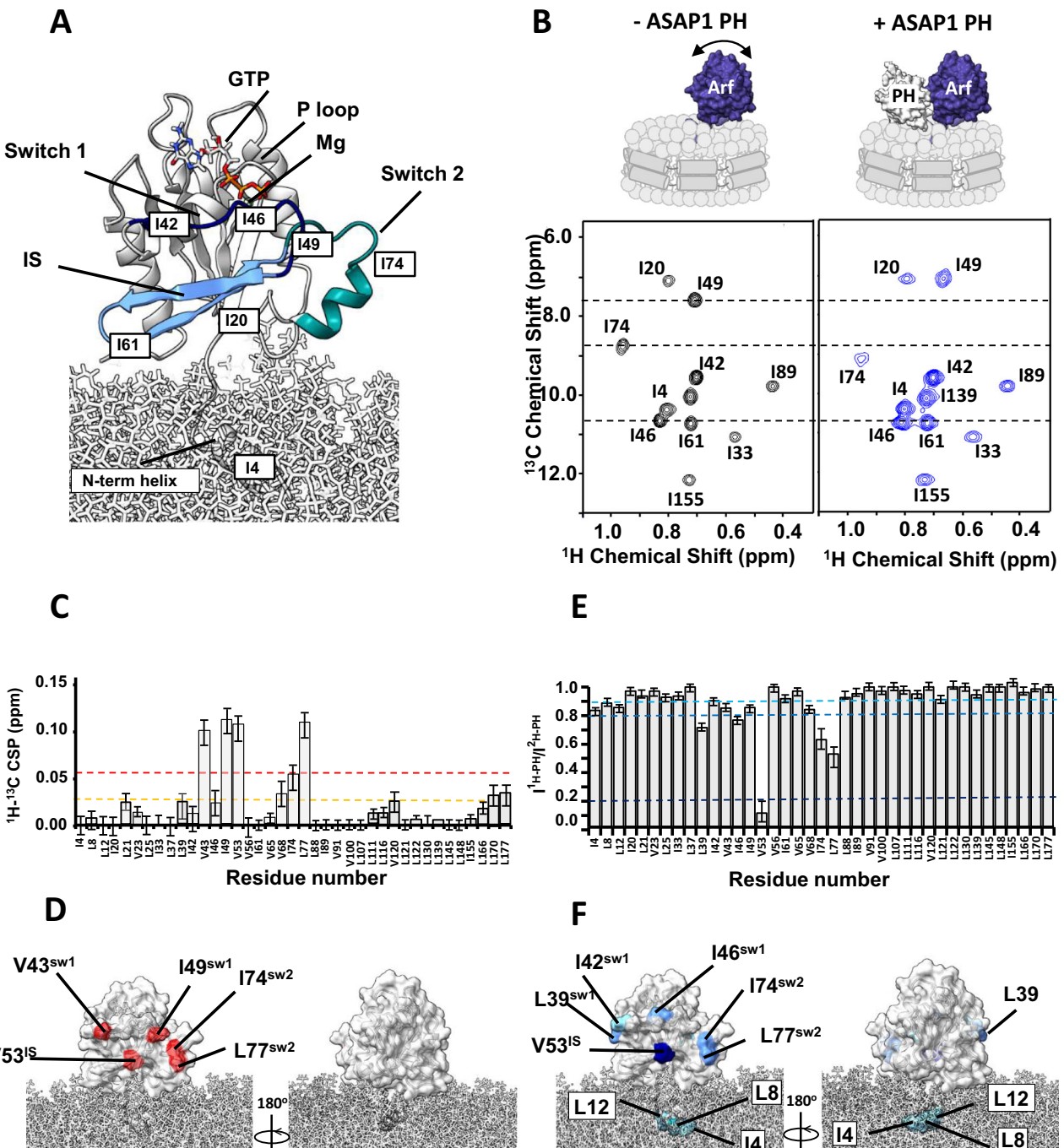

**Fig. 2 | Binding of ASAP1 PH to myrArf1 at the membrane. A** Homology model of human myrArf1[18] is shown in grey ribbon format at the membrane surface. Switch 1 (residues 40–49, dark blue), switch 2 (residues 68–78, cyan) and the interswitch (residues 50–67, light blue) are highlighted. Residues 2–13 form the N-terminal helix (embedded in the membrane), and residues 17–181 constitute the G domain (solvent-exposed). Images created using Chimera[56]. **B** $^1$H-$^{13}$C HMQC spectrum centered on the Ile region of 50 µM U-$^2$H,$^{15}$N, δ1-$^{13}$C$^1$H-labeled Ile, δ1/δ2 -$^{13}$C$^1$H-labeled Leu and γ1/γ2-$^{13}$C$^1$H-labeled Val myrArf1·GTP on a nanodisc (PC:PI(4,5) P$_2$ = 95:5) in the absence (left, black) or in the presence of $^2$H ASAP1 PH (right, blue) at a Arf:PH ratio of 1:1.2). **C** $^1$H-$^{13}$C CSPs observed between myrArf1 free or bound to ASAP1 PH domain (Arf:PH 1:1.2) plotted against residue number. When applicable, data are presented as the average of the pro-R and pro-S methyl groups. The error bars were calculated based on the digital resolution of the spectra, as described in

"Methods". **D** $^1$H-$^{13}$C CSP values are mapped on the Arf1 surface: CSP > 2σ (red), CSP > 1σ (orange). Images created using Chimera[56]. **E** Ratio of intensities of Arf1 $^1$H-$^{13}$C methyl cross peaks measured in the presence of $^1$H ASAP1 and $^2$H ASAP1 PH plotted against residue number. When applicable, data are presented as the average of the pro-R and pro-S methyl groups. Error bars were calculated based on the signal-to-noise (S/N) ratio of the spectra as described in Methods. **F** Ratios are mapped on the Arf1 surface with I$^{1H·PH}$/I$^{2H·PH}$ < 0.8 (dark blue) and 0.9 < I$^{1H·PH}$/I$^{2H·PH}$ < 0.8 (light blue). In addition to strong effects observed for V53 and residues of switch 1 and 2, the small amount of broadening observed for methyl groups on the N-terminal of myrArf1 may indicate proximity of the membrane bound part of myrArf1 with the ASAP1 PH domain as proposed in Roy et al.[4]. Images created using Chimera[56]. Source data are provided as a Source data file.

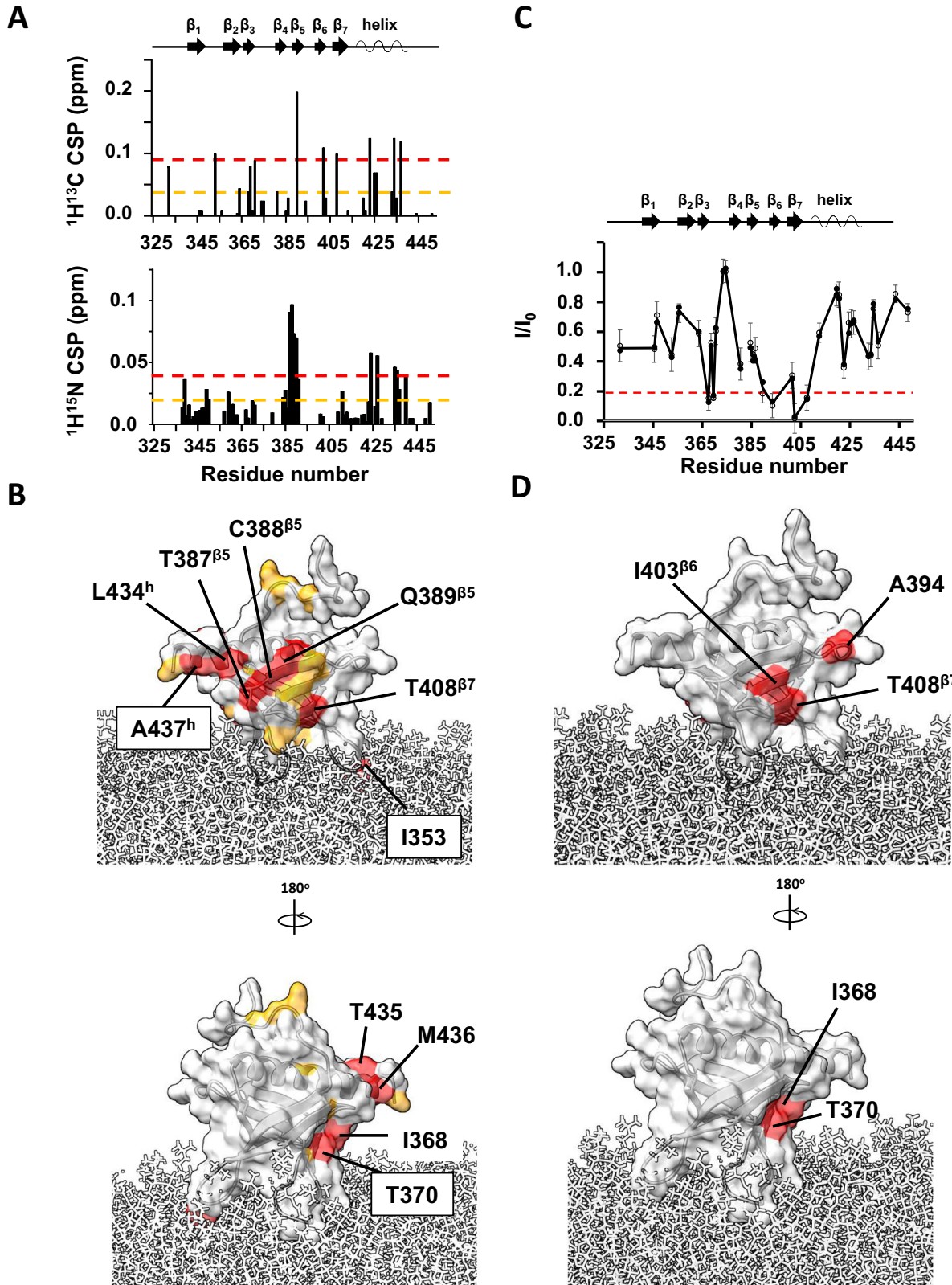

data, suggesting two states in dynamic equilibrium. Therefore, we chose to build 2 different sets of unambiguous distance restraints based on their location. Set$^{\beta_5/\beta_7}$ and Set$^{\beta_2/\beta_3}$ were built with PRE ratio <0.2 observed on the $\beta_5/\beta_7$ patch and $\beta_2/\beta_3$ patch. Both sets were complemented by ambiguous interaction restraints based on CSPs (Table SI2). We utilized those restraints in docking calculations to obtain models of the myrArf1:ASAP1 PH complex (Clusters$^{\beta_5/\beta_7}$ and

Clusters$^{\beta_2/\beta_3}$). We then used representative members of each cluster as starting conformations for multi-replica, multi-μs all-atoms MD simulations in the presence of PC: PI(4,5)P$_2$ membranes (SIM$^{\beta_5/\beta_7}$ and SIM$^{\beta_2/\beta_3}$). $^1$H-transverse relaxation for the *wt* PH domain methyls were calculated every ns and averaged over of the length of SIM$^{\beta_5/\beta_7}$ and SIM$^{\beta_2/\beta_3}$. We then compared the population weighted back-calculated PRE to the experimental ones (see material and methods and SI). We

**Fig. 3 | Binding of myrArf1 to ASAP1 PH at the membrane. A** $^1$H-$^{13}$C (top) and $^1$H-$^{15}$N CSPs (bottom) observed between ASAP1 PH bound to ND only or ND in the presence of myrArf1 (PH:Arf1 1:1.2) plotted against residue number. The error bars were calculated based on the digital resolution of the spectra, as described in "Methods". **B** CSP values are mapped on the Arf1 surface: CSP > 2σ (red), CSP > 1σ (orange). In addition to the large CSPs observed on the β sandwich, the large CSP observed for Ile353 (β$_1$/β$_2$ loop) could result from a reorientation of *wt* PH at the membrane in the presence of myrArf1 or from a direct interaction with myrArf1. Images created using Chimera[56]. **C** Intermolecular PRE profile measured on $^{13}$C methyl labeled ASAP1 PH in the presence of MTSL-tagged myrArf1K38C at the membrane surface. Two independent experiments were performed. All data points are plotted as open or filled circles. Error bars were calculated based on the signal-to-noise (S/N) ratio of the spectra as described in Methods. **D** PRE < 0.2 in the presence of MTSL-tagged myrArf1K38C mapped on the ASAP1 structure. Two patches were observed on opposite side of the β sandwich: The first patch includes residues Val390 (β$_5$ strand), Ala394 (β$_5$/β$_6$ loop), Leu402, Ile403 ((β$_6$ strand) and Ala413 on β$_7$ strand (called hereafter β$_5$/β$_7$ patch). The second patch includes residues Ile368 and Thr370 on the β$_3$ strand (called thereafter β$_2$/β$_3$ patch). Images created using Chimera[56]. Source data are provided as a Source data file.

found that back-calculated PREs correlated best with experimental PREs with a ratio β$_5$/β$_7$: β$_2$/β$_3$ of ~9:1 (slope of 1.03 and Pearson's coefficient of 0.95), indicating that the interface predominantly involves the state wherein the β$_5$/β$_7$ side of ASAP1 PH interacts with myrArf1 with a buried surface area of ~1100 Å$^2$ (Figs. 4A, SI4C, D and SI5).

Seven intermolecular hydrogen bonds/salt bridges are consistently detected in the MD trajectories. Three involve Arf1 side chains, Glu41 (Switch 1) and Glu54 (interswitch (IS)), that form salt bridges with the sidechain of Lys391 (strand β$_5$) on ASAP1 PH and Asn52 (IS) that forms transient salt bridges with either Lys430 (C-terminal helix) or Gln389 (strand β$_5$) of the PH domain. Hydrogen bonds of the Phe51 and Val53 main chains (myrArf1) with Lys430 or Glu431 and Gln389 side chains of ASAP1 PH, respectively, provide additional stability. These interactions are supplemented by hydrophobic clusters between the IS and switch 2 regions of Arf1 and the β$_4$/β$_5$, β$_6$/β$_7$ loops and the end of the C-terminal helix of ASAP1 PH. Interestingly, transient hydrogens bonds are also detected between residues of the N-terminal extension of ASAP1 PH (Gly325, Thr327, Met329, His330, Gln331) and switch 1 of myrArf1 (Thr44, Thr45 and Ile46). Interfacial contacts in Arf are illustrated in Fig. 4B, upper panel and in the PH domain in Fig. 4B, lower panel. Key residues at the interface are illustrated in Fig. 4C.

## The ZA domain has a minimal contribution to wt PZA binding to myrArf1·GTP

We first tested the hypothesis that binding of the PH domain to Arf could affect the energetics of ZA binding. A Cys159-MTSL tagged myrArf1 was used to measure intermolecular PREs between myrArf1 (bound to a non-hydrolysable GTP analog), either alone or in complex with *wt* PH, and the isolated, isotopically labeled ZA domain, at the membrane surface of NDs. The absence of intermolecular PREs (100 μM of Arf or Arf: PH and 100 μM of ZA domain) indicated that less than 5% of the ZA domain was bound to Arf under those conditions, consistent with a K$_d$ for ZA·Arf in the 0.1- 1.0 millimolar range and little or no effect of *wt* PH on the K$_d$ (Fig. SI6). This was complemented by monitoring $^1$H-$^{15}$N NMR signals in TROSY-HSQC[16] of domain segmentally labeled PZA[17] (only the ZA domain was isotopically labeled) bound to membrane-bound myrArf1 and conducted at 25 °C. Contrary to Arf, for which the orientational space is reduced when bound to the PH domain (manifest by the reduction in signal, compare Fig. 5B(a) to (b)[18]), the conformational space explored by the ZA domain is not reduced in the presence of the Arf: PH complex (no change in signal, comparing Fig. 5A(a) to (b)), as would have been expected from a high affinity complex between ZA and Arf1 (Fig. 5A, B). Taken together, this shows that binding of the PH domain to Arf results in minimal, if any, conformational coupling that would significantly alter the weak affinity of the ZA domain for Arf.

## Binding of the PH domain to Arf remodels the GTP binding pocket

The most significant conformational change involves switch 1, where residues 40–45 move outward, while residues 48–50 move inward, toward the GTP, with Pro47 acting as a hinge point (Figs. 6A and SI7A). This triggers the reorganization of polar residues within the nucleotide

binding site. Of particular relevance, we found that Thr48 and Lys30 are positioned in closer proximity to the Mg$^{2+}$ ion and the γ-phosphate of GTP, respectively (Figs. 6B and SI7B). The latter leads to an increase in the probability of the sidechain NH$_3$-group of Lys30 to establish an H-bond with the γ- phosphate oxygens (+25%) (Fig. 6B). The complex formation is linked to the stabilization of switch 2 in Arf, as indicated by average RMSD values for residues 71–80 of switch 2 of 3.8 Å for Arf alone and 3.0 Å for Arf in the presence of *wt* PH (illustrated as diameter of the tubes in Fig. 6A). Notably, switch 1 movement and switch 2 stabilization are key conformational changes that define activation of RAS protein GTPase activity[19,20].

Because several changes involved residues near the nucleotide-binding site, we examined the effects on the bound nucleotide that accompanied complex formation using naturally occurring $^{31}$P resonances of GTPγS bound to myrArf1 alone or in complex with ASAP1 PH. As observed previously[21], only one set of resonance lines is observable at 25 °C indicating that myrArf1·GTPγS exists predominantly in a single (NMR-distinguishable) conformational state at the membrane (Fig. 6C, left). The addition of the wt PH domain triggers deshielding of the γ-phosphate resonance, indicating a less negatively charged γ-phosphate upon Arf binding to ASAP1 PH (Fig. 6C, left inset and cartoon). A lower charge may facilitate the nucleophilic attack of water necessary for GTP hydrolysis and is consistent with changes of polar residues around the GTP when Arf is complexed with PH observed computationally (Fig. 6B).

Simulations also showed that when the complex forms, Glu54 in Arf1 shifts from intramolecular interactions to forming a salt bridge with Lys391 in the PH domain, contributing to the outward motion of switch 1, a conformation stabilized by the N-terminal extension of the PH domain (Figs. 7A and SI7C). Inferences from the MD simulations were tested using NMR. Altering the length of the N-terminal extension of ASAP1 PH to form $^{ΔN14}$PH specifically affects Val43 and Ile46 of switch 1, indicating proximity between residues 325-334 of *wt* PH and switch 1 of Arf (Fig. 7B) in agreement with the simulations. In addition, the spectral perturbation for Val43 was abolished by either randomizing the N-terminal extension amino acid sequence or mutating Tyr327 to Ala and was reduced by mutating His330 and/or Gln331 to Ala (Fig. SI7D). This indicated that (i) the tyrosine ring caused a ring current effect that altered the chemical shifts of Val43, indicating proximity between the two, and (ii) the interaction with switch 1 is sequence dependent. We also found that switch 1 residues were less affected by protonation/deuteration when bound to $^{ΔN14}$PH than to *wt* PH (Fig. SI7E), consistent with the loss of transient contacts between the N-terminus and switch 1, when myrArf1 is complexed to a shortened version of the PH domain.

Consistent with this, $^{ΔN14}$PH, lacking residues interacting with Arf switch 1, showed 100-fold less activity than PH despite near-wild-type membrane binding and only 2-fold weaker Arf binding (Fig. SI8 and Table SI1). Similarly, Lys391Ala in the PH domain and Glu54Ala in Arf had 100- and 70-fold less GAP activity, while binding to Arf was reduced by only 2-fold compared to wild type protein and binding to PI(4,5)P$_2$-containing membranes was unaffected by the Lys391Ala substitution (Fig. SI8E), consistent with Lys391 having a crucial role in

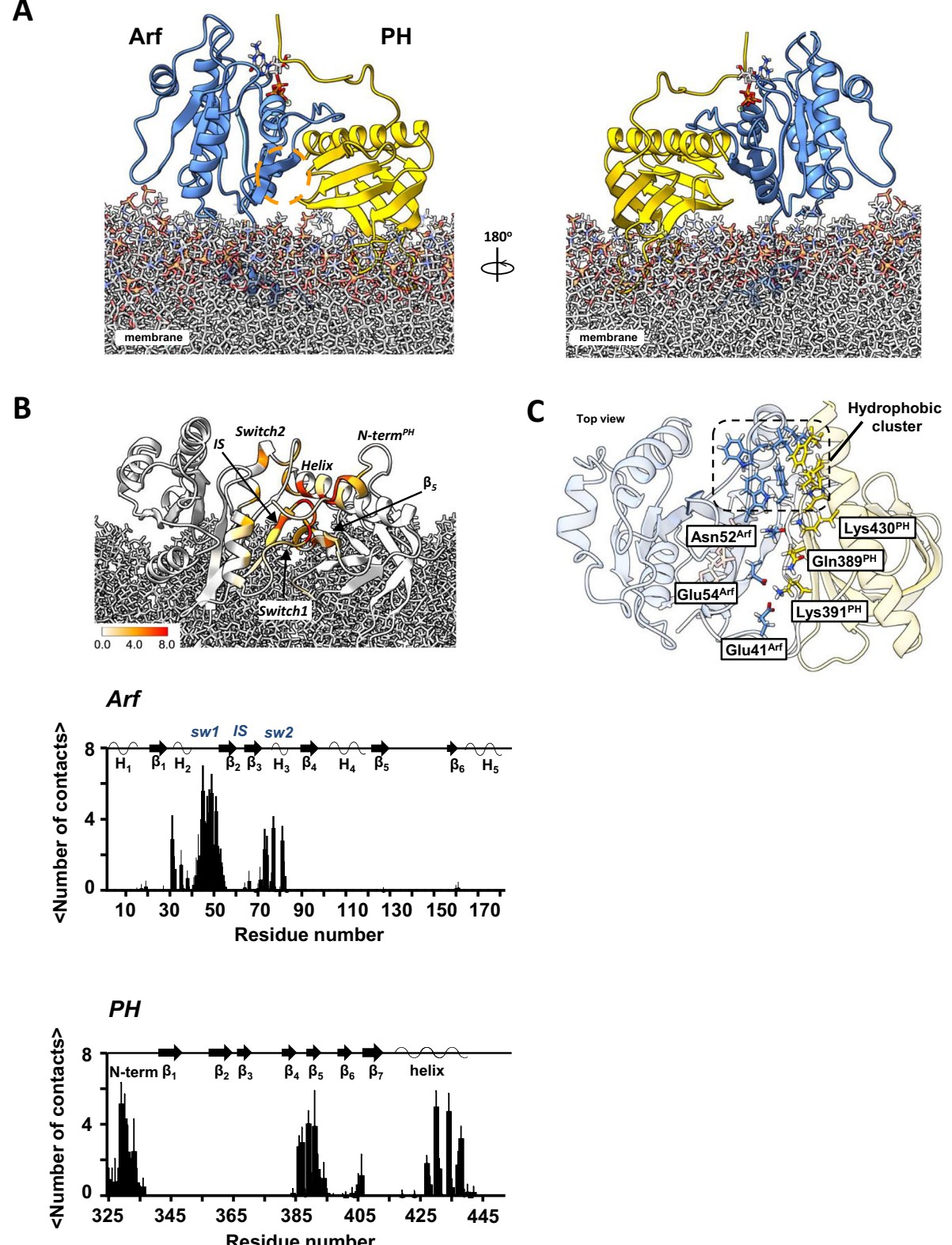

**Fig. 4 | Model of ASAP1 PH:myrArf1 complex at the membrane surface.**
**A** Snapshot representative of the model of the complex between myrArf1 (blue ribbon) with the GTP nucleotide represented as ball and stick and ASAP1 PH (gold ribbon) at the membrane surface. **B** ASAP1 PH: Arf interaction counts plotted against residue number and plotted on the model. Data were averaged over all MD trajectories with a cutoff of 3.5 Å to detect nonhydrogen proximities. **C** Key interactions at the interface. Stable intermolecular salt bridges and residues forming the hydrophobic cluster comprising Phe51, Trp66, Ile74, Leu77 and Trp78 on Arf1 and Leu 386 (β₄/β₅ loop), His405 (β₆/β₇ loop) and Leu 434, Ala 437 and Phe438 (end of the C-terminal helix) on ASAP1 PH are shown. All images created using Chimera[56]. Source data are provided as a Source data file.

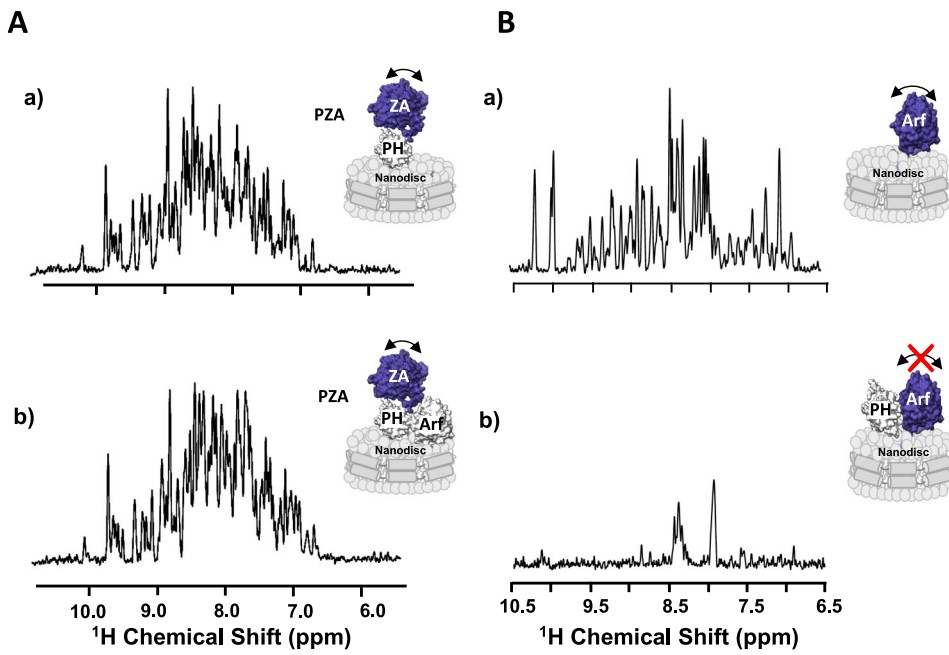

**Fig. 5 | The ZA domain binds weakly to the myrArf1:PH complex. A** One-dimensional $^1$H NMR spectrum corresponding to the projection of an $^1$H-$^{15}$N TROSY-HSQC along the F1 dimension obtained for ZA domain labeled PZA bound to ND alone (a) or bound to Arf (b), measured at the membrane surface of NDs containing PI(4,5)P$_2$. The absence of visible changes indicates that the ZA domain reorients independently from the complex between Arf, PH and the ND, as expected from weak affinity between ZA and the ARF:PH complex. Motional reorientation is represented by a double-sided arrow. Isotopically labeled, NMR visible domain used in each experiment is colored in blue. **B** One-dimensional $^1$H NMR spectrum corresponding to the projection of an $^1$H-$^{15}$N TROSY-HSQC along the F1 dimension obtained for myrArf1 bound to ND alone (a), in the presence of *wt* PH (b).The addition of a stoichiometric ratio of ASAP1-PH resulted in the loss of nearly all amide backbone resonances in the $^1$H-$^{15}$N TROSY-HSQC spectrum, in stark contrast with the spectrum in the absence of the PH domain indicating that the G domain becomes locked, reorienting with the same correlation time as the nanodisc. Motional reorientation is represented by a double-sided arrow. Isotopically labeled, NMR visible domain used in each experiment is colored in blue. Source data are provided as a Source data file.

allostery. In contrast, mutations at interface residues His405 and Phe438 showed activity reductions (5- and 15-fold) more proportional to their expected binding effects. Glu441 mutations, predicted to be outside the interface, had no effect (Figs. 7C, SI9 and Tables SI3–SI5).

Taken together, these data suggest that contacts between ASAP1 PH and myrArf1 alter the conformation and dynamics of functionally relevant regions around the nucleotide binding pocket compatible with the Arf:PH complex at the membrane surface being in an activated state primed for catalysis. Two elements on each protein—the N-terminal extension and Lys391 on the PH domain, and switch 1 and Glu54 on Arf1—are key motifs that mediate most of the activation upon PH binding to Arf.

**Binding of PH to Arf boosts the catalytic rate to a level comparable to membrane recruitment**

To quantify the relative contribution of membrane recruitment, binding to Arf1 and allosteric activation, we modeled the reactions of PH, ZA and Arf1 for the *"in trans"* with a system of 18 nonlinear ordinary differential equations (ODEs) and the *"in tandem"* with a system of 20 ODEs (see SI for details)[22–24]. A simultaneous fit to the data (Figs. 8A and SI10) indicated that, when considered independently, membrane recruitment and substrate binding by PH each increase GAP activity by three orders of magnitude compared to ZA alone. When combined, membrane recruitment and substrate binding account for only 4 orders of magnitude of the enhanced GAP activity as formation of the Arf-PZA complex reaches its maximal value and $k_{cat}$ becomes the rate-limiting step. A 10,000-fold increase in the catalytic rate $k_{cat}$ accounts for the remaining three to four of the roughly eight orders of magnitude increase in GAP activity observed experimentally between ZA alone and *wt* PZA in bilayers containing PI(4,5)P2 (Fig. 8B and

Tables SI5 and SI7). The increase in $k_{cat}$ is consistent with conformational changes located around the nucleotide binding site of myrArf1 observed in simulations and by NMR. The role of the PH domain in the reaction scheme is summarized in Fig. 8C. The PH domain is recruited to the membrane, increasing the likelihood of encounter with Arf to which it binds. The PH domain then induces a conformational change in Arf, leading to an activated state primed for GTP hydrolysis when the catalytic arginine, originating from the ZA domain, is introduced, thus the increase in $k_{cat}$. Based on these fits, we cannot exclude an additional factor(s) that explain the relatively minor deviation of the experimental and fitted data for PZA in PIP2, which are the focus of additional studies.

## Discussion

We defined the interface between myrArf1 and the ASAP1 PH domain on the membrane, allowing us to determine the structural basis and mechanistic underpinnings for enhancement of catalytic activity. The results indicate that the PH domain accelerates hydrolysis of GTP bound to Arf at the membrane surface via three distinct mechanisms. First, the PH domain serves as a phosphoinositide binding module to translocate ASAP1 PZA to the membrane where it searches a restricted volume for its substrate, increasing collision frequency and thereby decreasing time to collision. Second, the PH domain acts as the primary binding site for its substrate Arf·GTP, the physical link to the Arf GAP domain providing proximity to the catalytic arginine in the Arf GAP domain and DxxGQ glutamine in the switch 2 region of Arf1. Third, the formation of the complex alters Arf intramolecular H-bond network at the interface and triggers the remodeling of the nucleotide binding site in Arf·GTP to form an activated state of Arf, where GTP is primed for hydrolysis (Fig. 8C). For ASAP1 PZA, the contribution of

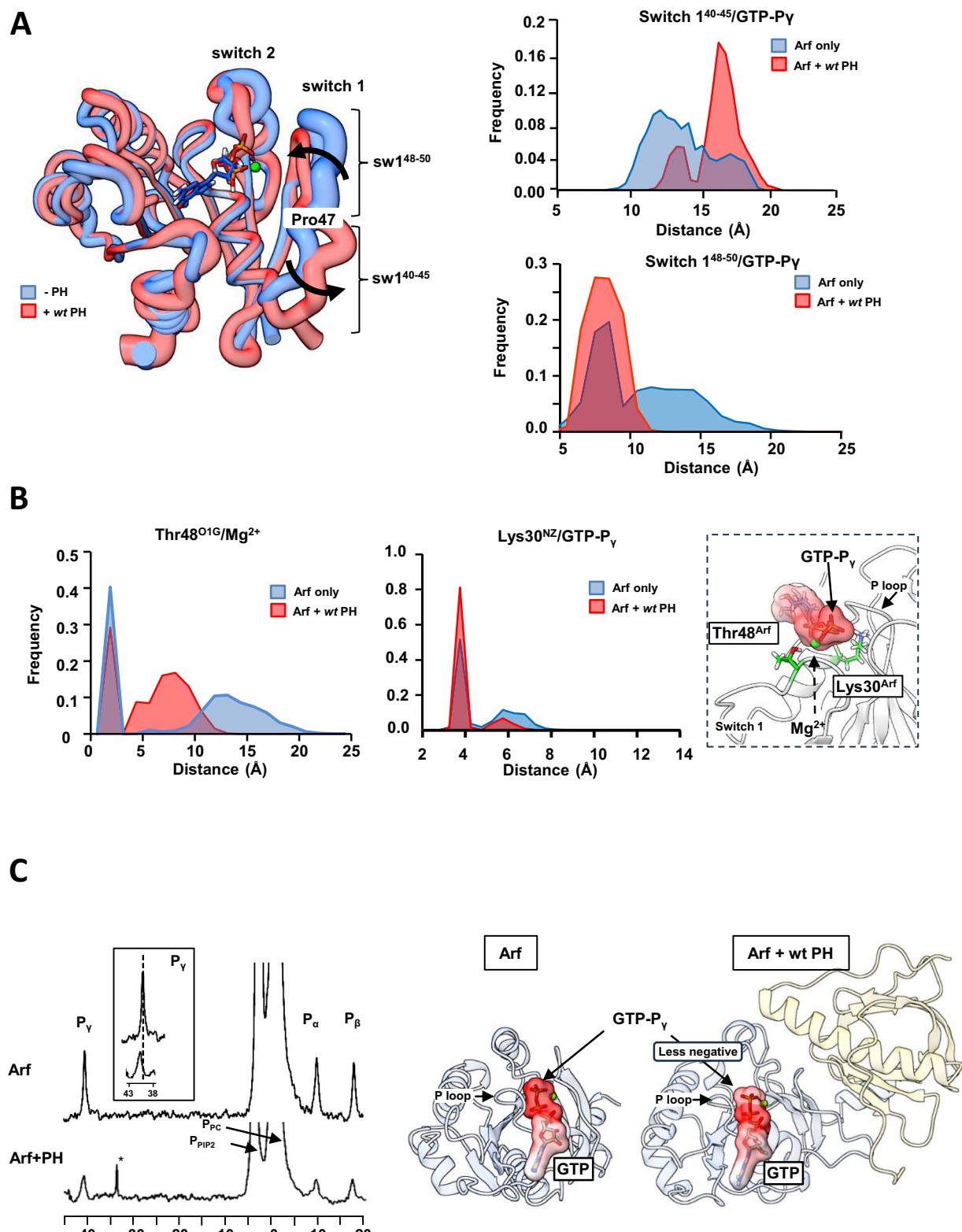

substrate binding and remodeling of the GTP binding pocket are of similar magnitude and together are greater than the contribution of membrane recruitment. In this model, the PH domain is integral to catalysis. PH domains in other Arf GAPs, as well as other regulators of the Ras superfamily, might function similarly. The results are relevant to Arf signaling in that, at least in this case, the GAP domain is not a complete GAP and the substrate binding site resides in a separate

domain. Thus, in ASAP1, the best substrate for the GAP domain is not Arf but Arf bound to the PH domain.

The interface between Arf and the PH domain is greater than 1000 $\text{Å}^2$ and, when bound to Arf1, the ASAP1 PH domain disrupts the intramolecular H-bond network that occurs in the small GTPase. Experiments and computations suggest essential roles of internal H-bond networks in shaping protein conformational dynamics[25,26]. For

**Fig. 6 | The PH domain alters Arf switch 1 conformation. A** (left) Cartoon representation of the conformation of switch 1 for Arf (blue) or Arf+ wt PH (red). The average structure calculated over the entire length of the simulation is represented. The thickness of the worm is proportional to the RMSD. (right) Histogram of the distance distribution between COM of residue 40–45 (top) or residue 47–50 (bottom) of switch 1 and GTP-Pγ for Arf (blue) and Arf + $wt$ PH (red) calculated over the entire length of the simulations. **B** (left) Histogram of the distance distribution between γ-OH atom of Thr48 and $Mg^{2+}$ for Arf (blue) and Arf + $wt$ PH (red) calculated over the entire length of the simulations. (middle) Histogram of the distance distribution between ζ-N of Lys30 and GTP-Pγ for Arf (blue) and Arf + $wt$ PH (red) calculated over the entire length of the simulations. (right) Cartoon representation of the position of Thr48 and Lys30 of Arf near GTP-Pγ as observed in MD simulations. **C** (left) $^{31}P$ NMR spectra of Arf bound to GTPγS in the presence (bottom) and absence of ASAP1 PH (top). Phosphates observed are labeled by phosphate types (P-α,β,γ: GTP; $P_{PIP2,\ PC}$: Lipid headgroup). The inset shows deshielding of GTP-Pγ chemical shift in the presence of ASAP1 PH. *, trace γ-$P_i$. (right) Cartoon representation of the electrostatic charge variation in GTP as observed by $^{31}P$ NMR. GTP-Pγ is less negatively charged when Arf is bound to ASAP1 PH, facilitating the nucleophilic attack of water. All images created using Chimera[56]. Source data are provided as a Source data file.

instance, in G Protein-Coupled Receptors, an extensive hydrogen bond network that spans all functional motifs of the protein in the active state has been identified, suggesting it might contribute to their activation mechanism. Here, the PH domain establishes competing salt bridges with Arf residues that are critical to the organization of the nucleotide binding site primed for GTP hydrolysis. Most of the changes in salt bridges are mediated by key motifs: the N-terminal extension and Lys391 on the PH domain, and switch 1 and Glu54 on the interswitch on Arf1 (Fig. 7). The result is a conformational state with some, but not all characteristics of the transition intermediates observed in crystal structures of members of the Ras superfamily in complex with their GAP proteins[27]. In particular, while switch 2 conformation is stabilized in the complex, we did not observe major changes in the conformational space explored by Arf1 Gln71 (Gln61 in RAS), a key catalytic residue in the RAS superfamily[28,29]. It is likely that the GAP domain, which is absent here, would affect Gln71 conformation as shown for ASAP3[30].

The interface between ASAP1 PH and Arf1·GTP differs from the three other Arf·PH domain complexes reported in the literature. These structures include crystal structures of a complex between N-terminally truncated Arf1 or Arf6 and the Arf binding domain of ARHGAP21[31] or [247-399]Grp1[32], and an NMR derived model of the interaction between yeast Arf1 and the PH domain of four-phosphate-adaptor protein 1 (FAPP1) at the membrane[33]. When overlaying Arf1 from our model to Arf1 in the FAPP1, ARHGAP21 or Grp1 complexes we observed that although all PH domains engage with a similar surface of Arf1, our model is the only one where strands β5/7 of PH interact with the interswitch of Arf1 at the membrane.

The catalytic mechanism we defined here is likely relevant to other Arf GAPs with PH domains N-terminal to the GAP domain. A structural alignment of the PH domains of ACAP1, AGAP1, and ARAP1 with that of ASAP1 reveals that Lys391 in the PH domain of ASAP1, crucial for allosteric regulation, is conserved among those ARFGAP families within a conserved XVK motif (Fig. 9). This finding suggests that the allosteric mechanism could be maintained across the ARFGAP family, despite widely different protein sequences. The idea that Arf GAPs use Arf in complex with other proteins may extend to other Arf GAPs. One example is Arf GAP2, with little or no detectable activity with isolated Arf. Activity is increased by 2 to 3 orders of magnitude by coatomer[34]. Cryo-EM studies supported the idea that Arf GAP2 was attacking Arf bound to coatomer[35].

The shift in mechanistic paradigm we propose provides a rationale for developing inhibitors for proteins with PH domains occurring in tandem with a catalytic domain. When the prevailing paradigm for PH domain function was membrane recruitment, efforts to develop inhibitors focused on blocking phosphoinositide binding to prevent recruitment, which has been described for Akt[36–40]. However, inhibitors targeting the PH domain of the Rho guanine exchange factor (GEF) pREX and Brag2 inhibit activity without blocking membrane association[41,42]. NAV-2729 binds to the PH domain of ASAP1, near one PI(4,5)$P_2$ binding site[43]. Neither PI(4,5)$P_2$ binding, nor membrane association are affected but GAP activity is inhibited. We speculate that the inhibitory small molecules affecting pREX and Arf exchange factors

and GAPs might alter a conformation in the PH domain that is necessary for catalytic activity. Additional structural characterization and biochemical studies will be valuable for defining the inhibitory mechanisms.

While defining the role of the PH domain is a critical step towards understanding the catalytic mechanism of Arf GAPs, our understanding is still incomplete. For instance, we cannot explain why the measured $k_{cat}$ is 10-fold greater than can be accounted for by the three contributions to catalysis that we describe. In current studies toward determining the full catalytic mechanism, we are examining two protein elements. First, the GAP domain-Arf interface is of obvious importance to catalytic mechanism. Second, the linker between the PH and Arf GAP domain may contribute to activity beyond restricting spatial volume. We cannot exclude additional factors. For instance, while we found that acid lipid composition in our model membranes was critical for activity and activity was otherwise independent of lipid composition, the model membranes may not fully capture the complexity and compositional diversity of native cellular membranes. As for the work presented here, addressing these questions will require an integrated structural biology and biochemical approach.

In summary, the PH domain of ASAP1 is an integral element of the catalytic GAP site. Identifying this function is essential for understanding the regulation and function of Arfs and should also be considered for other small GTPases.

## Methods

### Protein expression and purification

**Expression and purification of myrArf1 and mutants.** The human Arf1 (Uniprot P84077) gene was cloned into MCS1 of the pETDuet-1 vector (Novagen) between *Nco l* and *Sac I* restriction sites. A GSGSHHHHHH-tag was added at the C-terminus of human Arf1. The yeast NMT (Uniprot P14743) gene was subcloned into MCS2 of the pETDuet-1 vector between *Ndel I* and *Xho I* restriction sites.

For expression of the myr-Arf1 protein ( ~23 kDa) for NMR studies, the pETDuet-1 plasmid was transformed into *E. coli* BL21 Star (DE3) cells (Invitrogen), plated on LB agar plate containing carbenicillin (100 mg/L) for overnight growth. For expression of non-isotopically labeled protein, the freshly transformed colonies were picked, resuspended into 50 mL of Luria-Bertani medium containing carbenicillin (Goldbio, C-103-50) and incubated in a shaking incubator at 37 °C until an $OD_{600}$ of about 0.6. The culture was then diluted one-to-one with 50 mL of fresh Luria-Bertani medium. After repeating the same dilution procedure twice, culture was then diluted one-to-four with 800 mL of fresh Luria-Bertani medium and incubated until an $OD_{600}$ of about 1 before being diluted one to one and moved to a shaking incubator pre equilibrated at 22 °C (for a final culture volume of 2 L). For the production of myr-Arf1, sodium myristate (Sigma-Aldrich, M8005) was added 10 min before induction to a final concentration of 100 μM and 50 mg/L coenzyme A sodium salt (Sigma-Aldrich, C3144) to promote efficient N-myristoylation. Protein expression was induced at an $OD_{600}$ of 0.8 by adding isopropyl-β-D-thiogalactopyranoside (IPTG, Goldbio, I248C50) to a final concentration of 0.2 mM. For expression of U-[$^2H$, $^{15}N$], $^{13}CH_3$ methyl labeled protein, the freshly transformed colonies

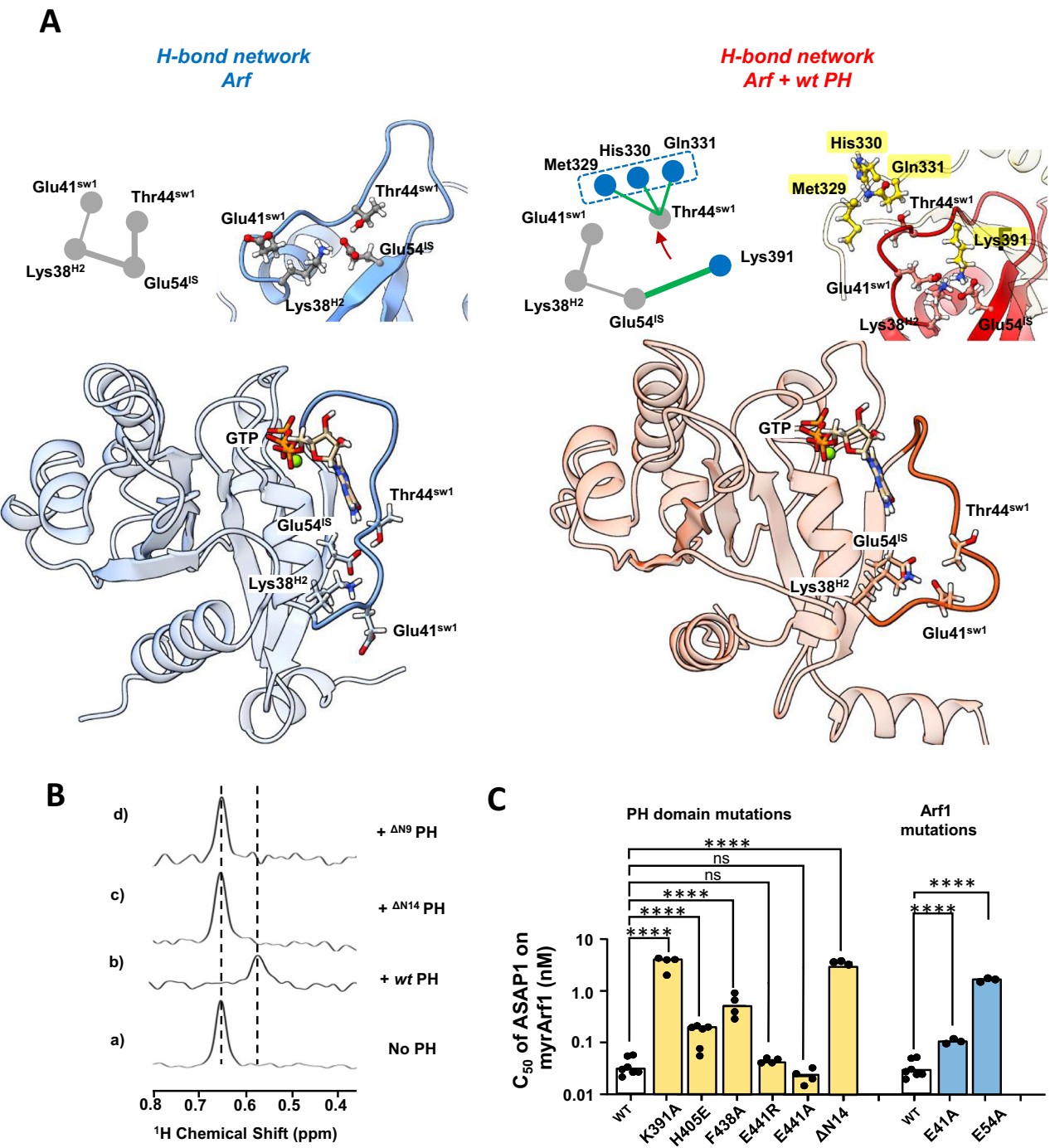

**Fig. 7 | Arf1 internal hydrogen bond network is destabilized in the myrArf1:PH complex. A** (Top) Changes in the hydrogen bond network linking Glu54, Thr44 and Lys38 in Arf1 (left) upon interaction with *wt* PH (right) highlighting the centrality of Glu54. H-bond network within Arf (grey line) or to PH (green line). Thickness of the line indicates the propensity of the H-bond. (Bottom) Cartoon depiction of Arf illustrating the spatial arrangement of key residues and the interplay between the hydrogen bond network and conformational changes in switch 1. Images created using Chimera[56]. **B** Stack of rows extracted from a ¹H-¹³C HMQC experiment along the proton dimension of Val43 (myrArf1) in the absence (a) or in the presence of *wt* ASAP1 PH (b), ᐃN14ASAP1 PH (c) or ᐃN9ASAP1 PH (d). **C** Effect on GAP activity assays of selected mutations on Arf1 (blue) and ASAP1 PH (yellow). C50 values (the concentration of PZA required to achieve 50% of GTP hydrolysis in 3 min) from each independent experiment are shown with $4 \leq n \leq 7$. Error bars represent standard deviation. ns, not significant with $p = 0.8251$ and $0.4128$ for the E441R and E441A mutants, respectively; ****, $p < 0.00001$ via one-way ANOVA with repeated measures (and mixed effects) and Dunnett's multiple comparisons test against WT. Source data are provided as a Source data file.

were picked and resuspended into 10 mL of M9/H₂O media for overnight growth at 37 °C in a shaking incubator. Then the overnight culture was poured into a fresh M9/H₂O media with a total volume of 5 mL and OD₆₀₀ of 0.2 and continued to grow until an OD₆₀₀ of about 0.6. The culture was diluted one-to-one with M9/D₂O medium (prepared with D-[²H;¹²C]-glucose, Cambridge Isotope Laboratories, Inc; DLM-2062-10 and ¹⁵N ammonium chloride, Sigma-Aldrich, 299251) and incubated until OD₆₀₀ reached 0.6. After repeating the same dilution procedure twice (with the final culture volume of 40 mL), the cells were spun down and resuspended in 200 mL M9/D₂O medium for

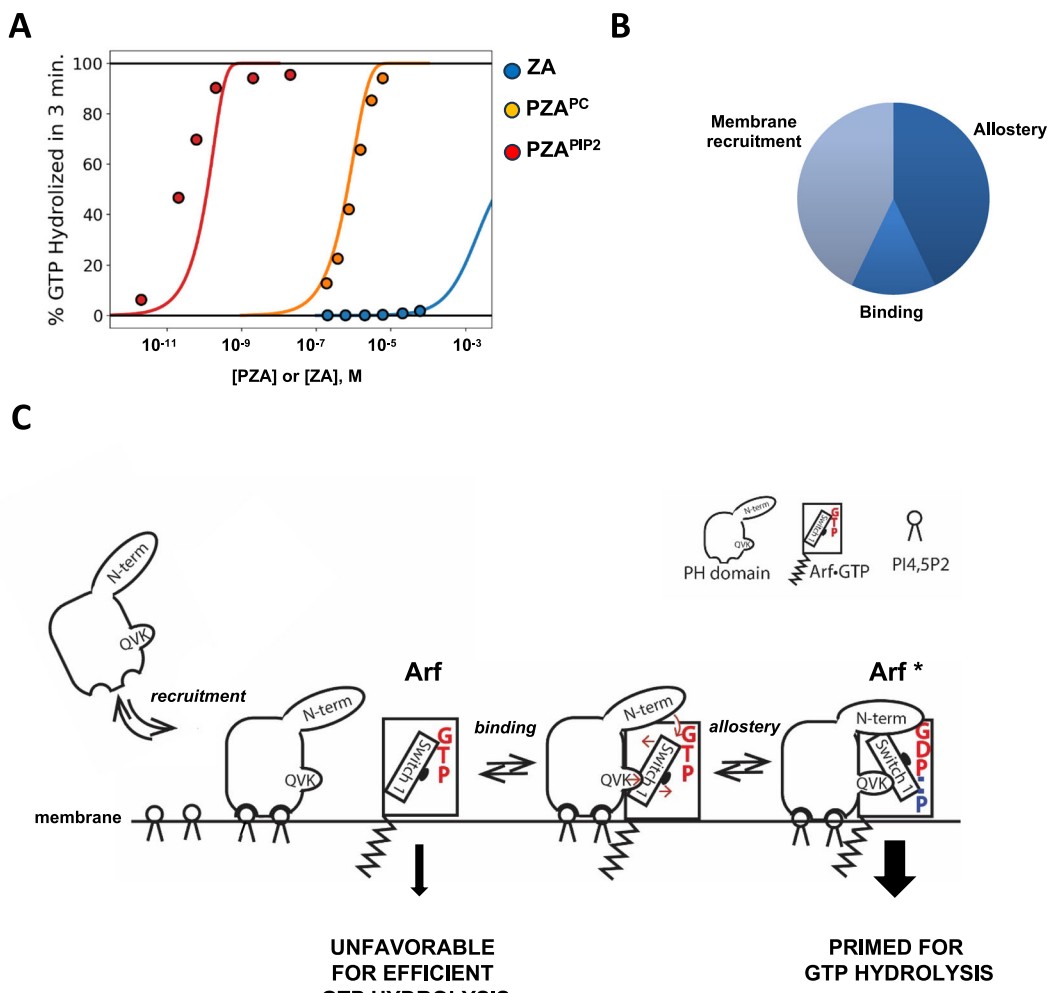

**Fig. 8 | Proposed mechanism of Arf regulation by ASAP1 PH domain. A** Kinetic modeling: Fraction of GTP hydrolyzed in 3 min by either ZA or PZA. From right to left, solid curves are for ZA (blue), PZA without PI(4,5)P$_2$ (orange), and PZA with PI(4,5)P$_2$ (red). Points represent experimental data and lines are predictions from optimized kinetic ODE model. In all cases where PI(4,5)P$_2$ is present its concentration is 5 mol%. **B** Relative contribution of membrane recruitment, binding and allostery to GAP activity by the ASAP1 PH domain. **C** Proposed mechanism of Arf regulation by ASAP1 PH domain. The PH domain binds PI(4,5)P$_2$ on the membrane (recruitment), forms a complex with Arf (binding), and induces motion of switch 1 controlled by Lys391 (part of XVK motif, see Fig. 9), and the PH domain N terminal extension—that results in a reduction in charge density on the γ phosphate such that Arf bound to ASAP1 PH is primed for efficient GTP hydrolysis. Source data are provided as a Source data file.

overnight growth at 37 °C[44]. The expression culture was made from the overnight culture by diluting to a volume of 2 liters with a starting OD$_{600}$ of about 0.2. After the culture had reached an OD$_{600}$ of 0.6, the temperature was reduced from 37 °C to 22 °C. For the production of myr-Arf1, sodium myristate (Sigma-Aldrich, M8005) was added 10 min before induction to a final concentration of 100 µM. At the same time, the media was supplemented with: 1) 50 mg/L 2-keto-3-[D$_2$],4-[$^{13}$C]-butyrate (Cambridge Isotope Laboratories, Inc. CDLM-7318) and 100 mg/L 2-keto-3-[D]-[$^{13}$CH$_3$,$^{12}$CD$_3$]-isovalerate (Cambridge Isotope Laboratories, Inc. CDLM-7317) to enable selective labeling of ILV methyl groups; 2) 50 mg/L coenzyme A sodium salt (Sigma-Aldrich, C3144) to promote efficient N-myristoylation. Protein expression was induced at an OD$_{600}$ of 0.8 by adding IPTG to a final concentration of 0.2 mM. The culture was incubated for additional 16 h at 22 °C for protein expression.

Cells were harvested by centrifugation at 7,000 g, 4 °C for 30 min. The cell pellets were resuspended in 25 mL lysis buffer (20 mM Tris-HCl, pH 8.0, 150 mM NaCl, 20 mM imidazole, 1 mM MgCl$_2$, and 0.5 mM tris(2-carboxyethyl) phosphine (TCEP) with one tablet of EDTA-free protease inhibitor (Thermo Scientist, A32965). The cells were lysed with a model 110S microfluidizer (Microfluidics) and clarified by

centrifugation at 48,000 g and 4 °C for 45 min. The lysate was loaded onto two 5 mL HisTrap HP columns (GE Healthcare). After the columns were washed with six column volumes (CVs) of lysis buffer, Arf1 and myr-Arf1 were eluted with a buffer containing 300 mM imidazole and 250 mM NaCl in a linear gradient from 20 mM to 300 mM imidazole over 14 CVs. The purity of myr-Arf1 was examined by LC-MS. The fractions containing purified myr-Arf1 were pooled and kept at 4 °C for further processing. The fractions containing both Arf1 and myr-Arf1 were combined and concentrated to a volume of one milliliter. Sodium chloride crystals were added to the sample to a final concentration of 3 M. After centrifugation at 21,000 × g and 4 °C for 15 min, the supernatant was collected and applied to a 5 mL pre-equilibrated HiTrap Phenyl HP hydrophobic interaction column (GE Healthcare) using a running buffer with 20 mM Tris-HCl, pH 7.4, 3 M NaCl, 1 mM MgCl$_2$, and 0.5 mM TCEP. After the column was washed with ten column volumes of running buffer, myr-Arf1 was eluted with 20 mM Tris-HCl (pH 7.4), 1 mM MgCl$_2$, and 0.5 mM TCEP using a linear gradient. The purity of myr-Arf1 was confirmed by LC-MS. Combining with previously purified myr-Arf1, the myr-Arf1 was exchanged to a buffer condition of 20 mM Tris-HCl, pH 7.4, 150 mM NaCl, 1 mM MgCl$_2$, and 0.5 mM TCEP, concentrated to about 100 µM, and stored at −80 °C for

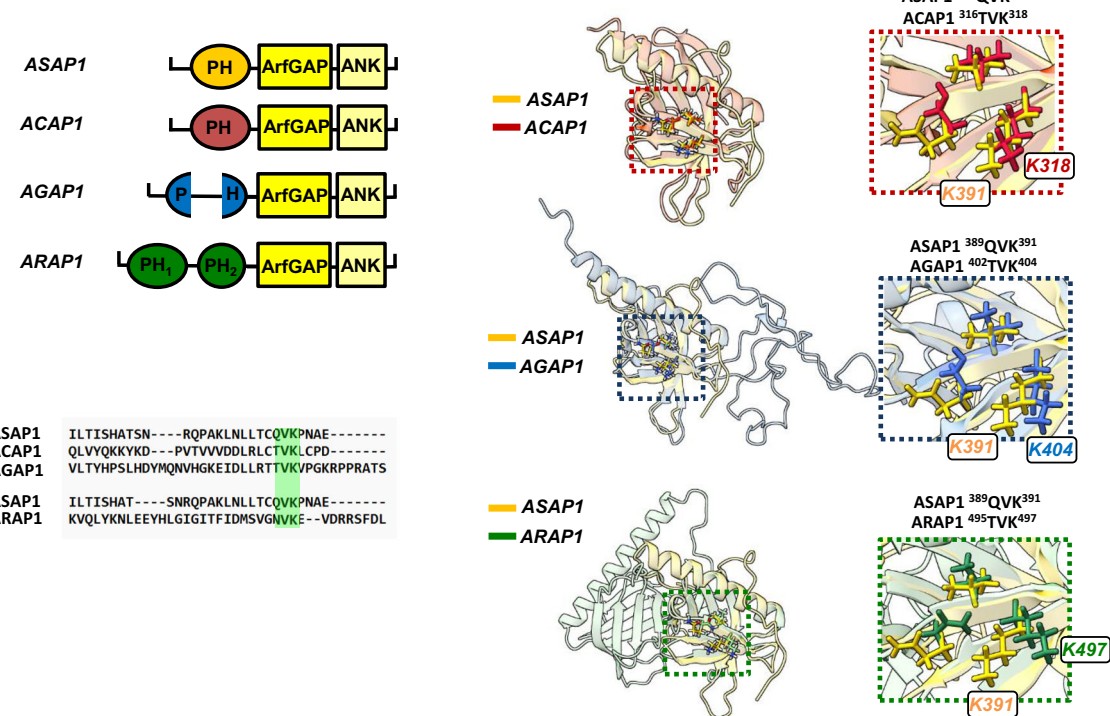

**Fig. 9 | The XVK motif is conserved in other ArfGAPs.** (Top left) Schematic organization of ASAP, ACAP, AGAP and ARAP family PZA domains. (right) Structural alignment of ASAP1 PH (yellow) with ACAP1 (red), AGAP1 (blue) and ARAP1 (green) PH domains highlighting the conservation of the XVK motif in each family. (bottom left) Structural alignment of their amino acid sequences centered around the XVK motif. All images created using Chimera[56]. Source data are provided as a Source data file.

further usages. The protein concentration was calculated by measuring the absorbance at 280 nm using a molar extinction coefficient of 29,450 $M^{-1}$ $cm^{-1}$.

Mutants of myrArf1 were generated by site-directed mutagenesis of the template plasmid (human Arf1 and yeast NMT in pETDuet-1, described above) using custom DNA oligos (Integrated DNA Technologies). Briefly, for each individual reaction 25 ng of template DNA was used with 125 ng of each custom oligo to perform 18 cycles of mutagenesis using a high-fidelity DNA polymerase (PfuTurbo, Agilent). Afterward, the mutagenesis reactions were subjected to *DpnI* treatment (New England Biolabs) to remove non-mutated DNA. Each transformation reaction was then transformed in NEB® 5-alpha competent *E. coli* (New England Biolabs) according to manufacturer's protocol and plated onto LB/Carbenicillin plates. Individual colonies were grown up in liquid media supplemented with Carbenicillin, after which the cell pellets were miniprepped (Qiagen) to isolate plasmid DNA. Clones were verified by Sanger sequencing with the pET Upstream primer (Novagen) at the NIH Center for Cancer Research (CCR) Genomics Core at the National Cancer Institute, Bethesda, MD. Colonies with desired Arf1 mutations (and no changes in yeast NMT) were then transformed into BL-21 Star (DE3) *E. coli* and used for expression and purification as listed above.

For expression of myArf1 and mutant proteins for binding and kinetic studies, the pETDuet-1 plasmid was transformed into *E. coli* BL21 (DE3) cells (Invitrogen), plated on LB agar plate containing ampicillin (100 mg/L) for overnight growth. Subsequent steps were as described above. Structural integrity of the Arf mutants was assessed by determining binding to GTPγS[35S] (ref).

**Preparation of wt ASAP1 PH domain and mutants.** The sequence for mouse ASAP1 PH domains, [325-451]–ASAP1, [334-451] –ASAP1 and [339-451]–ASAP1 (~14 kDa), was cloned between Nde I and Bam HI restriction sites of the pET3a vector, which was then transformed into

*Escherichia coli* BL21 Star (DE3) cells (Invitrogen) for protein expression, plated on LB agar containing carbenicillin, and incubated overnight (o/n). After, the seed cultures were used to inoculate large-scale (1 L) cultures, and the cultures were grown until the OD600 reached ~0.6 – 0.8 at which point they were induced with 1 mM IPTG for three hours at 37 °C. Following induction, the cells were harvested by centrifugation, supernatant removed, and frozen at −80 °C until purification.

PH domains were purified by resuspending in buffer A (50 mM Tris pH 7.4, 150 mM NaCl) by using 30 mL buffer for 1 liter's worth of cell pellet. A single protease inhibitor tablet (cOmplete EDTA-free, Roche) was used for each cell pellet. The cells were lysed using a cell disruptor (Microfluidics) and then ultracentrifuged (48,000 g for 30 min) at 4 °C. Afterward, the supernatant containing PH domain was removed, and all subsequent purification steps were conducted at room temperature as it was observed that the PH domain precipitates when chilled. The supernatant was applied to a 5 mL HiTrap SP HP column (Millipore Sigma) pre-equilibrated with buffer A, washed with 10 column volumes (CVs) of buffer A, then eluted with a 6 CV linear gradient from buffer A to buffer B (50 mM Tris pH 7.4, 1 M NaCl). Eluates containing PH domain were pooled and then injected onto a ~120 mL HiPrep 16/60 Sephacryl S-100 HR SEC column (Millipore Sigma) pre-equilibrated with buffer A supplemented with 1 mM TCEP. SEC eluates containing PH were then pooled, concentrated using 3000 MWCO spin-concentrators (Amicon), and snap-frozen using liquid nitrogen. Concentration of PH domain was determined by ultraviolet (UV) spectroscopy ($\varepsilon^{280} = 16,960$ M − 1 $cm^{-1}$).

For the production of [U-²H], [U-¹⁵N]-methyl specifically labeled protein, $NH_4Cl$ is substituted by ammonium chloride (¹⁵N ≥ 99%), D-glucose is replaced by d-(²H, ¹²C)-glucose (²H ≥ 98%), and ¹³$CH_3$-methyl specifically labeled precursors are added as described below. For a typical cell culture of 500 ml, a few freshly transformed colonies of BL21 Star (DE3) cells were picked to inoculate 5 ml of M9/$H_2O$

minimal media for o/n growth at 37 °C in a shaking incubator (250 rpm). One milliliter of the o/n culture [typical optical density at 600 nm (OD600) ~1.2] was then used to inoculate 4 ml of fresh M9/ H$_2$O medium to achieve a starting OD600 of 0.25. At OD600 ~ 0.5, 5 ml of M9/D$_2$O minimal media was added and cell growth continued until an OD600 of ~0.5 is reached. Cells were diluted again by a factor of 2 and growth followed to OD600 ~ 0.5. This cycle was repeated until a D$_2$O/H$_2$O ratio of 3:1 (20 ml total) is reached. Cells were then harvested by centrifugation (3000 $g$ for 30 min) and resuspended in 25 ml of M9/ D$_2$O, and growth was continued in a 100-ml baffled flask until an OD600 of 0.5 is reached, before an additional 25 ml of M9/D$_2$O was added for o/n growth at 37 °C. When the o/n OD600 was between 1.3 and 1.5, the o/n cell expression (50 ml) was added to 500 ml of M9/D$_2$O and growth followed at 37 °C, up to OD600 ~ 0.6. For selective δ1-$^{13}$C$^1$H-labeled Ile, δ1 -$^{13}$C$^1$H-labeled Leu, γ1-$^{13}$C$^1$H-labeled Val, β -$^{13}$C$^1$H-labeled Ala and γ2-$^{13}$C$^1$H-labeled Thr labeling, the PLAM-AβIγ1LV$_{proS}$Tγ kit was used (NMR-Bio). After the addition of the precursor according to the manufacturer's protocol, cell growth continued until an OD600 of approximately 0.8 at 20 °C is reached, at which time protein expression was induced with the addition of 1 mM IPTG. After induction, another 2 g/liter of D-($^2$H, $^{12}$C)-glucose was added, and the culture was grown o/n at 20 °C. All subsequent steps were carried as described above.

For expression of ASAP1 PH and mutant proteins for binding and kinetic studies, the pET21 vector plasmid was transformed into *E. coli* BL21 (DE3) cells (Invitrogen), plated on LB agar plate containing ampicillin (100 mg/L) for overnight growth. Subsequent steps were as described above.

**Expression and purification of PZA and mutants.** The PH, Arf GAP, and ankyrin repeats (PZA construct, residues 325–724) of ASAP1 were expressed and purified as previously described (PMID: 9819391). Briefly, PZA with a 10× N-terminal His tag was expressed by transforming into BL-21(DE3) competent cells, then picking individual colonies and growing using LB/ampicillin media until seed cultures were at an OD600 of ~0.6. After, the seed cultures were used to inoculate large-scale (1 L) cultures, and the cultures were grown until the OD600 reached ~0.6–0.8 at which point they were induced with 1 mM IPTG for three hours at 37 °C. Following induction, the cells were harvested by centrifugation, supernatant removed, and frozen at −80 °C until purification.

PZA was purified by resuspending in nickel buffer A (20 mM Tris pH 8, 500 mM NaCl, 20 mM imidazole) by using 30 mL buffer for 1 liter's worth of cell pellet. A single protease inhibitor (cOmplete EDTA-free, Roche) was used for each cell pellet. The cells were lysed using a cell disruptor (Microfluidics) and then ultracentrifuged at >100,000 $g$ for one hour at 4 °C. Afterward, the supernatant containing PZA was removed. The supernatant was applied to a 1 mL HisTrap HP column (Millipore Sigma) pre-equilibrated with nickel buffer A, washed with 10 column volumes (CVs) of nickel buffer A, then eluted with a 10 CV linear gradient from nickel buffer A to nickel buffer B (20 mM Tris pH 8, 500 mM NaCl, 500 mM imidazole). Eluates containing PZA were pooled and then injected onto a ~120 mL HiPrep 16/60 Sephacryl S-200 HR SEC column (Millipore Sigma) pre-equilibrated with storage buffer (20 mM Tris pH 8, 150 mM NaCl). SEC eluates containing PZA were then pooled, concentrated using 20,000 MWCO spin-concentrators (Amicon), and snap-frozen using liquid nitrogen.

Mutants of PZA were generated by site-directed mutagenesis of the template plasmid (PZA in pET-19b) using custom DNA oligos (Integrated DNA Technologies). Briefly, for each individual reaction 25 ng of template DNA was used with 125 ng of each custom oligo to perform 18 cycles of mutagenesis using a high-fidelity DNA polymerase (PfuTurbo, Agilent). Afterward, the mutagenesis reactions were subjected to DpnI treatment (New England Biolabs) to remove non-mutated DNA. Each transformation reaction was then transformed in NEB® 5-alpha competent E. coli (New England Biolabs) according to manufacturer's protocol and plated onto LB/ampicillin plates. Individual colonies were grown up in liquid media supplemented with ampicillin, after which the cell pellets were miniprepped (Qiagen) to isolate plasmid DNA. Clones were verified by Sanger sequencing with standard T7 forward and terminal primers at the NIH Center for Cancer Research (CCR) Genomics Core at the National Cancer Institute, Bethesda, MD. Colonies with desired PZA mutations were then transformed into BL-21(DE3) E. coli and used for expression and purification as listed above.

**Preparation of isotopically domain labeled PZA.** Domain labeled PZA was obtained using Sortase mediated ligation (SML) and purified as previously described with some modifications[17]. Briefly, the reaction buffer (SML buffer) was 50 mM Tris, pH 7.8, 5 mM CaCl2, 100 mM NaCl, 10 mM L-Arginine, 0.2 mM TCEP. The stoichiometry for the reaction was ZA: PH: Srt 1:2:0.25. Typical reaction volume was 5 mL, and the concentration of isotopically labeled ZA was set at 10 µM. The SML reaction was performed in a centrifugal concentrator [MWCO 3000 Da, Amicon] while spinning at 3000 g. After a given time (10 min), the volume of the reaction was readjusted to 5 mL with SML buffer and 10 µM of fresh PH domain was added. This was repeated 4 times for a total reaction of ~45 min. Sortase ligated PZA was purified in a single step over a 5-ml SP anion exchange column (GE Healthcare) to minimize protein degradation. The column was equilibrated and washed with buffer A (20 mM Tris pH 7.4, 100 mM NaCl), and the protein was eluted using a 20-column volume gradient to 50% buffer B (buffer A and 1 M NaCl). The domain labeled PZA was usually eluted at 38–42% buffer B. Buffer exchange into buffer A was done stepwise by successive centrifugation in a centrifugal concentrator [MWCO 3000 Da, Amicon] to minimize protein precipitation. The sample identity was confirmed by mass spectrometry, and purity was assessed by SDS–polyacrylamide gel electrophoresis. Concentration of domain labeled PZA domain was determined by ultraviolet (UV) spectroscopy ($\varepsilon^{280}$ = 36900 M$^{-1}$ cm$^{-1}$).

**Expression and purification of MSPΔH5.** The plasmid for MSPΔH5 was a gift from F. Hagn and G. Wagner (Harvard Medical School). The protein was expressed and purified as described previously[10].

**Dissociation rate of mantGTP measured by Fluorescence spectroscopy**

Binding and dissociation of mantGTP was monitored by following the FRET signal resulting from resonance energy transfer from Arf tryptophan to the methylanthronoyl (mant) group on GTP (excitation 290 nm; emission 448 nm) using a FluorMax3 spectrophotometer (Jobin Yvon Horiba, Edison, NJ). To load myrArf1 with mant-GTP (ThermoFisher, M12415), 1 µM myrArf1 was incubated at 30 °C for 40–60 min in 25 mM HEPES, pH 7.4, 100 mM NaCl, 1 mM dithiothreitol, 0.5 mM MgCl$_2$, 1 mM EDTA, 5 µM mant-GTP, and 0.5 mM Large Unilamellar Vesicles (LUVs) with 5% mol PI(4,5)P$_2$. After Mant-GTP loading reached plateau, different concentrations of ASAP1 PH were added to the reaction mixture and excess GTP (160 µM) was added to initiate the mant-GTP dissociation from Arf. Arf·mantGTP has a FRET signal, whereas Arf·GTP does not; therefore, the exchange results in a decrease in fluorescent signal.

**Preparation of myr-Arf1·GTPγS anchored on MSPΔH5 nanodiscs for NMR spectroscopy**

**Preparation of empty nanodiscs.** All lipids were purchased from Avanti Polar Lipids, Inc. To prepare nanodiscs, acyl chain perdeuterated 1,2-dimyristoyl-sn-glycero-3-phosphadidylcholine (DMPCd54) in chloroform solution (Sigma-Aldrich 319988) and 1,2-dioleoyl-*sn*-glycero-3-phospho-(1′-myo-inositol-4′,5′-bisphosphate)     (PI(4,5)P$_2$)     in chloroform: methanol (Millipore MX0488): water (20:9:1) solution

were mixed and air-dried with nitrogen flow before solubilization with cholate in aqueous buffer [20 mM tris-HCl (pH 7.4), 150 mM NaCl, and 75 mM sodium cholate (Millipore SX0420)]. Nanodiscs were assembled by mixing MSPΔH5 with solubilized lipids at a ratio of 1:45 (final cholate concentration of 18 mM) followed by the removal of cholate from the mixture with Bio-Beads SM2 resin (Bio-Rad, 152-8920), under o/n rocking at 22°. Assembled NDs were then purified via a Superdex-200 size exclusion column (GE Healthcare) and concentrated on a centrifugal concentrator [10 kDa molecular weight cutoff (MWCO), ThermoFisher Scientific]. The concentration of NDs was determined by UV spectroscopy ($\varepsilon^{280} = 18,450$ M $- 1$ cm$^{-1}$).

**Preparation of myr-Arf1·GTPγS anchored nanodiscs.** MyrArf1·GDP was incubated with freshly prepared nanodiscs in 20 mM Tris-HCl, pH 7.4, 150 mM NaCl, 0.5 mM MgCl$_2$, 0.5 mM TCEP, 1 mM EDTA, and 2 mM GTPγS (Millipore Sigma G8634). After incubation at room temperature for 30 min, myrArf1·GTPγS anchored on MSPΔH5 NDs was purified by a Superdex 200 Increase 10/300 GL column (GE Healthcare) into the final NMR buffer (20 mM Tris-HCl (pH 7.4), 150 mM NaCl, 1 mM MgCl$_2$, and 0.5 mM TCEP). Typically, the fractions containing nanodisc-anchored myr-Arf1·GTPγS were pooled, concentrated and analyzed by SDS-PAGE. Known concentrations of Arf and MSP were run on the same SDS-PAGE gel and used to determine the Arf1: nanodiscs ratio. Concentration of NDs was then adjusted such that the ratio between Arf and ND was ~1 by adding empty NDs.

**Generation of spin-labeled variants of myrArf1.** Spin-labeling conjugation reactions were performed on myrArf1·GTPγS anchored on nanodiscs purified in NMR buffer without reducing agent. The mono-cysteine variants of myrArf1 were treated with a 5-fold molar excess of S-(1-oxyl-2,2,5,5-tetramethyl-2,5-dihydro-1H-pyrrol-3-yl) methyl methanesulfonothioate (MTSL) (Toronto Research Chemicals Inc., 81213-52-7) and the reaction was allowed to proceed for 1 h at 25 °C in the dark. The unreacted spin-label was then removed by passing the protein samples through a Superdex 200 Increase 10/300 GL column pre-equilibrated in NMR buffer without reducing agent. The efficiency of the spin-labeling reactions was confirmed by LCMS mass spectrometry measurements. The reactions were found to proceed to completion in all cases.

### NMR spectroscopy
Experiments were performed using approximately 30–50 µM myr-Arf1(ASAP1 PH) anchored to 30–50 µM nanodiscs. Samples (approx. 250 µL) were contained in Shigemi microcells. Data were acquired at 25 °C or 32.5 °C using Bruker AVIII-850, AVIII-800 and AVIII-700 spectrometers equipped with cryogenic TCI probes. All NMR data were processed and analyzed using Topspin 3.6.4 and/or NMRPipe[45].

$^1$H,$^{13}$C HMCQ spectra of $^{13}$CH$_3$ methyl labeled proteins were acquired at 25 °C using a SO-FAST HMQC pulse sequence as implemented in NMRlib package[46]. The spectral widths were set to 12.94 and 25 ppm in the $^1$H and $^{13}$C dimensions, respectively and inter-scan delays were set to 1 sec. In total, 1542 × 256 complex points were recorded, and between 16 and 64 scans/FID gave rise to an acquisition time between 1.5 and 5 h. Prior to Fourier transformation, the data matrices were zero-filled to 4096 ($^1$H) × 1024 ($^{13}$C) complex points and multiplied by a cosine apodization function in both $^1$H and $^{13}$C dimensions.

$^1$H,$^{15}$N TROSY-HSQC spectra were acquired at 32.5 °C using the BEST TROSY principle as implemented in NMRlib package[46]. The spectral widths were set to 12.06 and 36 ppm in the $^1$H and $^{15}$N dimensions, respectively with a recycle delay set to 1.5 sec. In total, 1368 × 192 complex points were recorded, and 96–128 scans/FID gave rise to an acquisition time between 5 and 7 h. Prior to Fourier transformation, the data matrices were zero-filled to 4096 ($^1$H) × 1024 ($^{15}$N) complex points and multiplied by a cosine apodization function in both dimensions.

Chemical shift perturbations were calculated using Equation 1:

$$CSP_{H-X} = \sqrt{(\Delta\delta_H^2 + (A \cdot \Delta\delta_X^2)}$$ (1)

where $\Delta\delta_H$ is the change in amide or methyl proton value (in parts per million) and $\Delta\delta_X$ is the change in amide nitrogen or methyl carbon value (in parts per million). A is a scale factor equal to 0.17 (or 0.185) when X is $^{15}$N (or $^{13}$C).

For CSPs, the error values were calculated by the formula $[\Delta\delta^1 H \cdot R^1 H + \Delta\delta^{13} C \cdot R^{13} C(\text{or }^{15}N)/(5.88 \text{ or } 5.40)^2]/\Delta\delta$, where $R^1 H$ and $R^{13}C(\text{or }^{15}N)$ are the digital resolution in ppm in the $^1$H and $^{13}$C (or $^{15}$N) dimensions, respectively.

For comparison of Arf intensities in the presence of $^1$H or $^2$H PH domain, the error values were calculated by the formula in Eq. (2):

$$Error = \frac{I^2H}{I^1H} \sqrt{\left(\frac{1}{S/N}\right)_{2H}^2 + \left(\frac{1}{S/N}\right)_{1H}^2}$$ (2)

where $I^1H$, $(S/N)^1H$ and $I^2H$, $(S/N)^2H$ are the intensity and signal-noise ratios of resonance measured in the presence of protiated and deuterated PH domains, respectively.

1D-$^{31}$P NMR spectra were acquired at 25 °C on a Bruker 700 MHz NMR spectrometer with a 5 mm Prodigy broadband cryogenic probe using 70° flip angle pulses, ~15000 scans, an interscan delay of 7 s, an acquisition time of 84 ms, and a WALTZ-16 proton decoupling sequence.

**PRE NMR measurements.** Samples used for the paramagnetic relaxation enhancement (PRE) measurements contained 60 µM of protiated myrArf1 variants in their spin-labeled forms and 50 µM of U-$^2$H,$^{15}$N and δ1-$^{13}$C$^1$H-labeled Ile, δ1 -$^{13}$C$^1$H-labeled Leu and γ1-$^{13}$C$^1$H-labeled Val, β -$^{13}$C$^1$H-labeled Alanine (Ala) and γ2-$^{13}$C$^1$H-labeled Threonine (Thr) ASAP1 PH. For all spin-labeled variants, one sample each of the paramagnetic or the diamagnetic (in which the spin-label was reduced by incubation with 1 mM ascorbic acid for 2 h) species, were prepared. All measurements were performed at 25 °C. A recycle delay between scans of 4 s was used to insure adequate magnetization recovery for both the diamagnetic and paramagnetic states. The error values were calculated by the formula in Eq. (3):

$$Error = \frac{I}{I_0} \sqrt{\left(\frac{1}{S/N}\right)^2 + \left(\frac{1}{S/N}\right)_0^2}$$ (3)

where $I$, $(S/N)$ and $I_0$, $(S/N)_0$ are the intensity and signal-noise ratios of resonance measured in paramagnetic and diamagnetic samples, respectively.

### Docking
**Generation of ambiguous interaction restraints.** Ambiguous interaction restraints (AIRs) for use in the data-driven docking calculations (described below) were obtained for Arf1 and ASAP1 PH. Active residues were defined as those that were solvent exposed and displayed $^1$H-$^{13}$C or $^1$H-$^{15}$N CSPs > 0.1 ppm (i.e., >2σ). Passive residues were defined as solvent exposed residues within 6.5 Å of the active set and/or with methyl (amide) CSPs between 0.047 (0.02) and 0.095 (0.04) ppm (i.e., 1σ < CSPs < 2σ) for wt PH and methyl (amide) CSPs between 0.03 (0.02) and 0.06 (0.04) ppm for Arf. Residues were defined as solvent exposed if they displayed at least 20% of relative solvent accessibility. The AIRs used for the docking calculations are listed in Table SI2.

**Generation of unambiguous distance restraints from PREs.** The intensity ratio of the HMQC spectra before ($I$) and after reduction with ascorbic acid ($I_0$) were used to generate the unambiguous distance

restraints. Protons with $I/I_0 < 0.2$, including protons whose resonances were no longer detectable in the paramagnetic spectra, were assigned distance constraints ranging from 1.8 to 16 Å.

**Data-driven docking.** Structure calculations were performed using all ambiguous and unambiguous restraints using the HADDOCK version 2.2 suite[14,15]. The homology model of human myr-Arf1, generated based on yeast Arf1 as in ref. 18, was used as the starting structure for Arf after truncating residues 1–17. The starting structure for PH was the AlphaFold model of the protein[47]. Docking simulations consisted of three consecutive stages. In the first, rigid body docking and energy minimization stage, a total of 10,000 structures of the complex were calculated allowing each of structures from the Arf1 and ASAP1 PH ensembles to explore a sufficiently broad landscape of initial orientations. At the end of this stage, 400 structures with the lowest energy scores were selected for the simulated annealing (second) and water refinement (third) stages. Residues 40 to 49 of Arf1 (switch 1) and 325–332 (N-terminal extension), 352–355 ($\beta_1/\beta_2$ loop), 374–378 ($\beta_3/\beta_4$ loop) and 445–451 (C.terminus) of ASAP1 PH were kept fully flexible during this stage of molecular docking. For Set$^{\beta5-\beta7}$, 396 structures out of 400 were grouped into 1 major cluster. For Set$^{\beta2-\beta3}$, 287 structures were grouped into 3 major clusters, each comprising of at least 20 individual structures, representing about 70% of the 400 water-refined models generated after the final refinement stage. The most relevant clusters for each set (Clusters$^{\beta}_5/^{\beta}_7$ and Clusters$^{\beta}_2/^{\beta}_3$), as defined by the most favorable HADDOCK score (HS, $-128 \pm 2.7$ and $-113.1 \pm 0.8$), contained ~95% (396 of 400) and ~80% (213 out of 281) of the clustered structures, with an average RMSD value of $0.6 \pm 0.6$ Å and $2.6 \pm 0.4$ Å relative to the lowest energy structure.

## Molecular dynamic simulations

Because the initial docking stage was performed without membranes, we then used representative members of each cluster as starting conformations for all-atoms MD simulations.

Molecular systems for the starting point of simulations were built using CHARMM-GUI[48]. Relaxation using the standard steps provided by CHARMM-GUI was performed with NAMD. Production simulations employed the GPU optimized pmemd module of AMBER 18. The C36m forcefield was used for proteins[49] and lipids[50] with standard dynamics parameters (force-based switching between 10 and 12 Angstroms, SHAKE constraints on bonds to hydrogens, and the particle-mesh Ewald algorithm to handle long-ranged electrostatics with periodic boundary conditions). Zero tension at one atmosphere isotropic pressure was applied using a Monte Carlo barostat. Temperature was maintained at 310 K with a Langevin friction coefficient of 1 ps$^{-1}$. Five replicas of Arf, Arf:$wt$ PH (Clusters$^{\beta}_5/^{\beta}_7$) and Arf:$^{\Delta N14}$PH based on Arf:$wt$ PH (Clusters$^{\beta}_5/^{\beta}_7$) were run such that total simulation time for each system was at least two microseconds. Because we found that root mean square fluctuations (RMSF) between orientations over one replica was on the same order of magnitude as the RMSF between replicas for SIM $^{\beta}_5/^{\beta}_7$, only one replica was run for Arf:$wt$ PH Clusters$^{\beta}_2/^{\beta}_3$ (SIM$^{\beta}_2/^{\beta}_3$).

The simulated lipid bilayer was composed of DMPC (ca. 295 lipids) and approximately 5% PIP2 (ca. 7 each of DOPI24 and DOPI25, the dioleoyl PIP2 in the C36 forcefield with varied protonation of the PIP2 phosphate). The total water layer was approximately 7.5 nm high (ca. 25000 TIP3P water molecules). Initial coordinate and simulation input files and a coordinate file of the final output have been deposited at Zenodo (https://doi.org/10.5281/zenodo.16895013).

**Back calculation of PRE profiles from MD simulations.** PREs were computed using the RotamerConvolveMD (version 1.3.2) package employing MDAnalysis (v1.0.0[51],) as well as the rotamer library from[52]. The code was modified slightly to incorporate PREs using the target

hydrogens, as opposed to the default (the modified code files are available at http://github.com/alexsodt/premod). Briefly, atomic coordinates of MTSL and of $\delta$1-CH$_3$-Ile, $\delta$1/$\delta$2-CH$_3$-Leu and $\gamma$1/$\gamma$2-CH$_3$-Val of Arf1 were extracted from MD simulations (1 frame every ns for each of the five multi μs long simulations for a total of ~7500 frames) of Arf and Arf + wt PH and used to back calculate PRE rates. For each of the ~7500 frames, the effective PRE relaxation rate ($\Gamma_2^{cal}$) of a protein methyl proton was then calculated as the average over the PRE rate $\Gamma_{2,i}$ computed for each of the $N$ MTSL orientation the protein:

$$\Gamma_2^{cal} = \frac{1}{N} \sum_{i=1}^{N} \Gamma_{2,i} \qquad (4)$$

with $\Gamma_{2,i}$ equal to:

$$\Gamma_{2,i} = \kappa \left( 4\tau_c + \frac{3\tau_c}{1 + \omega_H^2 \tau_c^2} \right) r^{-6} \qquad (5)$$

with $\kappa$ equal to $1.23 \times 10^{-32}$ cm$^6$s$^{-2}$ for the proton spin as reported previously, $r$ the distance between the free electron and methyl group protons of Ile, Leu, Val, Ala or Thr residues in a single frame, $\tau_c$ the rotational correlation time of the electron-nuclear interaction, which was approximated using Eq. (6)

$$\frac{1}{\tau_c} = \frac{1}{\tau_R} + \frac{1}{\tau_S} \qquad (6)$$

where $\tau_R$ is protein rotational correlation time and $\tau_S$ is the electronic longitudinal relaxation time. We used a $\tau_R$ of 70 ns for Arf in complex with PH and a value of 100 ns for $\tau_S$. Calculation was then repeated for each frame and to back calculate average PRE rates for each methyl of SIM$^{\beta}_5/^{\beta}_7$ and SIM$^{\beta}_2/^{\beta}_3$.

Fit to the data was then performed as a population weighted average of the PRE rate for each pose of the PH domain (Eq. (7)).

$$\Gamma_{2,i} = \rho^{SIM\beta5/\beta7} \cdot \Gamma_{2,i}^{SIM\beta5/\beta7} + \rho^{SIM\beta2/\beta3} \cdot \Gamma_{2,i}^{SIM\beta2/\beta3} \qquad (7)$$

where $\Gamma_{2,i}$ is, $\rho^{SIM\beta6/\beta7}$ and $\rho^{SIM\beta2/\beta3}$ are the population of SIM$^{\beta6/\beta7}$ and SIM$^{\beta2/\beta3}$, respectively.

Then, the $\Gamma_{2,i}$ were converted to the intensity ratios of the paramagnetic (I) to diamagnetic (I$_0$) peaks for representation ($\frac{I}{I_0}$) using Eq. (8):

$$\frac{I}{I_0} = \frac{R_2 \times \exp\left(-\Gamma_2^{cal} \times t\right)}{R_2 + \Gamma_2^{cal}} \qquad (8)$$

where $t$ is the total evolution time of 6.89 ms in the HQMC pulse sequence, $R_2$ is the intrinsic transverse relaxation rate, which was estimated from the half-height line width of peaks in the diamagnetic spectra.

## Functional assays

**GDP-GTP exchange and GAP activity followed by Fluorescence spectroscopy on nanodiscs.** All fluorescence experiments measurements were performed with a Horiba Fluoromax spectrofluorometer in a 120 μL quartz cell. The sample (140 μL) was thermostated at 22 °C. The time constant of the fluorometer was set to 500 ms. The excitation wavelength ($\lambda_{exc}$) and emission wavelength ($\lambda_{em}$) were 297 and 337 nm, respectively. The excitation and emission bandwidth were set to 4 and 10 nm, respectively. Nucleotide exchange of purified myrArf1 (5 μM) was assessed by monitoring the change in tryptophan fluorescence following addition of Ethylenediaminetetraacetic acid (EDTA) (2 mM) in the presence of 20 μM of GTP and 500 μM of exposed lipids in

nanodiscs, which takes advantage of the difference in fluorescence between the GDP- and GTP-bound forms of Arf proteins. The reaction was stopped by the addition of 1 mM $MgCl_2$. Induction of hydrolysis of myrArf1·GTP to myrArf1·GDP was determined by following the change in tryptophan fluorescence, as previously described after addition of isolated ZA domain or *wt* PZA.

**GAP activity determined by hydrolysis of $[\alpha^{32}P]$GTP bound to Arf on LUVs.** LUVs were prepared by extrusion. Briefly, 1 μmol lipids (molar ratio, 40% PC, 25% PE, 15% PS, 10% cholesterol, and 10% total phosphoinositide) dissolved in chloroform, in a siliconized glass tube were dried under a nitrogen stream for 30 min to 1 h, followed by lyophilization for at least one hour. The dried lipids were resuspended in 200 μL 1× PBS, for a final concentration of 5 mM. The solution was vortexed, subjected to five rounds of freeze/thaw, and extruded using a lipid extruder (Avanti Polar Lipids) through a Whatman Nucleopore Track-Etched membrane with 1 μm pores. The LUVs were stored at 4 °C and were used within a week for activity assays.

GAP-induced conversion of myrArf1•GTP to myrArf1•GDP was determined as described previously[6,53–55]. Reaction mixtures contained 25 mM HEPES, pH 7.4, 100 mM NaCl, 1 mM dithiothreitol, 2 mM $MgCl_2$, 1 mM GTP, 0.5 mM LUVs, myrArf1 bound to $[\alpha^{32}P]$GTP, and variable concentrations of Arf GAP. The LUVs were included in the myrArf1 GTP loading reaction. The reactions were incubated at 30 °C for 3 min. (unless otherwise specified) and quenched with 2 mL of ice-cold 20 mM Tris, pH 8.0, 100 mM NaCl, 10 mM $MgCl_2$, and 1 mM dithiothreitol. Protein-bound nucleotide was trapped on nitrocellulose, and guanine nucleotide was released by addition of formic acid. $[\alpha^{32}P]$GDP and $[\alpha^{32}P]$GTP were then separated using thin-layer chromatography plates, and quantified.

### Mass spectrometry measurement
Mass spectra were obtained using Agilent Technologies 6100 Series Single Quadrupole LC/MS equipped with an electrospray source, operated in positive-ion mode. Separation was performed on a 300SB-C3 Poroshell column (2.1 mm × 75 mm; particle size 5 μm). The analytes were eluted at a flow rate of 1 ml/min with a 5 to 100% organic gradient over 5 min and holding the organic phase A for 1 min. Mobile phase A contained 5% acetic acid in water and mobile phase B was acetonitrile. Data acquisition, data analysis, and deconvolution of mass spectra were performed using Open Lab Chem Station Edition software (version C.01.05). Samples of purified proteins were typically 5 μl of a 5–10 μM solution.

### Reporting summary
Further information on research design is available in the Nature Portfolio Reporting Summary linked to this article.

## Data availability
PDB depositions referenced in this study are available from the PDB repository as follows: PDB: 3LVQ (ASAP3-Arf6 fusion construct comprised of residues 9–175 of Arf6 (Uniprot P62330) with residues 416–702 of ASAP3 (Uniprot Q8TDY4)); PDB: 2KSQ (Arf1 residues 2–181 (Uniprot P11076)); PDB: 2J59 (N-terminally truncated Arf1 (Uniprot P84077) with Arf binding domain of ARHGAP21 (Uniprot Q5T5U3); PDB: 4KAX (N-terminally truncated Arf6 with [247-399]Grp1) (Uniprot O43739). The PDB structure of the Arf1:FAPP1 PH domain (Uniprot Q9HB20) was graciously provided by Pr. James Prestegard. Source data are provided with this paper.

## Code availability
The molecular dynamics programs VMD version 1.9.4 and CHARMM36 force field have been described previously and inquiries should be addressed to A.J.S. The MD simulations were uploaded at Zenodo [Zenodo: 16895013].

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

## Acknowledgements

The research was supported by the Intramural Research Program of the National Cancer Institute (Projects ZIA BC 011419, ZIA BC 011131, and ZIA BC 011132 supported O.S., R.A.J., Y.Z., J.L., and R.A.B.; Project BC007365 supported X.J., E.M.R. and P.A.R.) and by the Intramural Research Program of the Eunice Kennedy Shriver National Institute of Child Health and Human Development (Project 1ZIAHD008955 supported A.J.S.). M.E.J. and S.L.F. gratefully acknowledge funding from an NIH MIRA Award R35GM133644 to M.E.J. F.H. was supported by the U.S. Department of Commerce, Award 70NANB17H299. The authors acknowledge the use of the Biophysics Resource, Center for Structural Biology, NCI, and the assistance of Dr. Sergey Tarasov and Ms. Marzena Dyba. The authors thank David Lambright, University of Massachusetts, and Michael Harris, University of Florida, and Vernon E. Anderson for insightful discussions and review of the manuscript. The content is solely the responsibility of the authors and does not represent the official views of the NIH. Certain commercial materials, equipment, and instruments are identified in this work to describe the experimental procedure as completely as possible. In no case does such identification imply a recommendation or endorsement by NIST, nor does it imply that the materials, equipment, or instrument identified are necessarily the best available for the purpose.

## Author contributions

O.S., S.L.F., X.J., M.E.J., A.J.S., P.A.R., and R.A.B. designed the research. O.S. performed NMR experiments, fluorescence experiments and

 

analyzed MD simulations. S.L.F. performed mathematical modeling. X.J. expressed and purified some of the mutants and determined their functional activities. J.L. performed protein biochemistry. R.A.J. prepared Arf1, various forms of ASAP1 PH and domain labeled ASAP1 PZA under the supervision of O.S. Y.Z. performed some of the initial NMR experiments. E.M.R. expressed some of the Arf mutants and determined their functional activity. B.J.H. analyzed some of the MD simulations under the supervision of O.S. and A.J.S. O.S., S.L.F., X.J., A.J.S., and P.A.R. analyzed the data and wrote the manuscript. O.S., P.A.R., F.H., M.E.J. and R.A.B. reviewed and edited the manuscript.

## Funding

## Competing interests

The authors declare no competing interests.
