## [Peer Review file · Nature Communications]

An Active Allosteric Mechanism in ASAP1-Mediated Arf1 GTP Hydrolysis

Corresponding Author: Dr Paul Randazzo

Version 0:

Reviewer comments:

Reviewer #1

(Remarks to the Author)

This manuscript nicely summarizes compelling data from NMR, biochemical, and computational analyses that, together, provide strong evidence for a direct interaction of the Arf1 GTPase with the PH domain of its GAP, ASAP1. The study demonstrates that this interaction results in conformational changes in the switch 1 region of Arf1 to prime GTP hydrolysis. That a GAP PH domain directly activates a GTPase is remarkable because its principal role was thought to be in membrane recruitment and related passive processes. The experiments, simulations, and their analyses are technically sophisticated and state-of-the-art, and the resulting data are internally consistent and, in general, provide a sound basis for the conclusions of the work. However, I have a few concerns that require attention before the manuscript can be accepted for publication.

1) One issue that requires careful consideration is the following: In my reading of the manuscript, the primary 'physical' interaction between ASAP1 and Arf1 on the membrane is through the PH domain, which also recruits the GAP to the membrane, leaving unclear the role of the ArfGAP (Z) domain which in other Ras-related GTPases is known to directly bind at the switch regions of the GTPases. My question then is, will the Z domain of ASAP1 directly engage Arf1 while the PH domain is still bound, or is this some sort of a sequential process? If the former, how is that possible considering that the two GAP domains will have to compete for the same binding region? The proposed model in Fig 8C is confusing in this context, because it is as though the Z domain is not required for "high GAP activity", which is obviously not the case since the PH domain lacks the critical Arg finger.

2) A related question involves the ~4-orders of magnitude difference in the GTP hydrolysis rate between the "tandem" and "in trans" systems (Fig 3E). This seems to suggest that the complexes of the tandem and in trans constructs with Arf1 is different and, if that is the case, the mechanism of rate enhancement could also be different. Therefore, how confident can one be about the specific interactions and conformational changes observed by studying the isolated PH domain with Arf1 in explaining the actual mechanism of action of ASAP1 in cells? An attempt is made to explain the different aspects of rate enhancement (membrane recruitment, allostery etc) using high-level mathematical modeling but the structure-mechanistic level explanation remains unconvincing.

3) Another concern, one that can potentially be addressed by a few control experiments and/simulations, is that all of the NMR, modeling and MD simulations of myr-Arf1 with or without PH were studied in DMPC bilayers. The choice of this lipid is curious because DMPC is not the most representative lipid species in mammalian cells. This requires justification or validation also because membrane biophysical properties including thickness, lipid packing and electrostatics are likely to modulate the observed intermolecular interactions. Moreover, while the use of simpler model membranes is understandable from a technical point of view, it is important to acknowledge that the actual lipid environment in which Arf1 resides (mostly in ednomembranes) is likely to affect many of the current observations, just like the presence of PIP2 alters membrane binding and catalysis.

4) It is stated in page 14 that "the substrate for the GAP domain is not Arf but Arf bound to the PH domain". This is a strong statement that may require justification beyond the current set of data. For example, are there prior studies supporting this notion, including from GTP hydrolysis experiments with isolated GAPs (more than one GAP) in solution or membrane?

5) There are a few grammatical errors: Examples: "average over of the"; page 8 -- remove 'of', "residues with the", page 10 -- replace "with" by 'within'; Fig 3 E is not labeled.

Reviewer #2

(Remarks to the Author)

In this interesting manuscript, Soubias et al. present a structural-mechanistic study of membrane recruitment and allosteric activation by the ASAP1 GAP-PH domain tandem, which mediates membrane recruitment and has GAP activity for Arf1. Quantitative biochemical analyses demonstrate large differences in GAP activity in the presence (and absence) of PIP2-containing liposomes for isolated domains (alone and in combination) vs. the GAP-PH domain tandem. The underlying structural/dynamic basis was thoroughly investigated by NMR in using nanodiscs containing PIP2 and MD simulations, which are used to identify interaction surfaces, construct plausible data-restrained docking models and investigate relevant functional dynamics. Key structural predictions are supported by structure-based mutational analyses. Perhaps the most salient structural-mechanistic insight relates to the observation that whereas most mutations in the PH domain-Arf1 interface have effects that correlate with effects on binding, one notable exception (involving Lys391 in the PH domain of ASAP1 and Glu54 in switch I of Arf1) strongly impairs the catalytic step with little effect on binding, thus supporting a new role for the PH domain in allosteric activation beyond established roles of PH domains in lipid/protein binding and membrane recruitment. Finally, similar relative contributions of membrane recruitment and allosteric activation to GAP activity are delineated using ODE-based mathematical modeling.

Overall, these studies represent an important conceptual advance with respect to the known functional roles of PH domains. The experiments, analyses and modeling appear to be quantitative, rigorous and thorough. The results are convincing and will be of interest to the Arf field as well as the GTPase and PH domain fields more broadly. Indeed, membrane recruitment vs. allosteric activation is relevant to many systems of biological interest but, as far as we are aware, this is the first case for which this important functional distinction is clearly delineated. The manuscript is generally well written, apart from a few instances noted below that would benefit from clarification.

Specific comments:

Does the 'in trans' (PH + ZA) experiment in Figure 1E necessarily separate membrane recruitment from other activation mechanisms? An allosteric activation mechanism requires that the PH domain interact with ZA. Is it also possible that this interaction might in principle enhance membrane recruitment of ZA, or perhaps both? In the absence of additional experiments (e.g., effect of mutations in the PH domain-ZA interface on membrane recruitment vs. allosteric activation) or perhaps ODE-based mathematical modeling, a more nuanced interpretation may be appropriate.

Have the authors considered the alternative possibility that the two patches on the PH domain identified by PREs might arise in the context of a higher order assembly with 2:1 stoichiometry rather than a two-state dynamic equilibrium in a complex with 1:1 stoichiometry? Would such a 2:1 complex (or mixture of 2:1 and 1:1) be consistent or not with the with data and docked models? For example, could both binding sites be occupied simultaneously or is binding mutually exclusive due to steric conflicts? Might be worth discussing briefly.

p. 11, "Notably, switch 1 movement and switch 2 stabilization are key conformational changes that define activation of RAS protein GTPase activity." Seems like a relevant citation(s) would be appropriate here.

Fig. 8A. Can the authors comment on the modest discrepancy between the fitted ODE models and the data for PZA-PIP2? Is it related mainly to errors for concentration estimation in pM range, details of the mathematical modeling/simultaneous fitting, or something else?

Assuming we recall correctly, ring current shifts are highly distance and orientation dependent. Is the ring current shift discussed in the last paragraph on p. 11 consistent with the structural model(s)?

Minor:

P. 5, "which both binds" should be "which both bind"

Fig. 1E, the label "E" is missing

Define acronyms on first usage (e.g., GEF, PC, etc)

There are a few other typos and inconsistencies throughout the manuscript. For example, some headings end with a period, other don't.

Reviewer #3

(Remarks to the Author)

Reviewer #4

(Remarks to the Author)

Arf1 GTPase regulates vesicular trafficking in cells. Arf1 has no GTPase activity on its own and instead requires binding to GTPase activating proteins (GAPs) to allow it to hydrolyze GTP to GDP. It was previously thought that GAP domains would be sufficient for the activation of Arf1. However, recent studies have demonstrated that the GAP domain of ASAP1 by itself is not sufficient to activate Arf1. This suggested that other domains in ASAP1 could play important roles in activating Arf1. However, it was not clear how these domains would help activate Arf1. The manuscript by Soubias et al. identifies that the membrane binding PH domain of ASAP1 is required for the activation of Arf1. Using NMR studies, the authors demonstrate that the PH domain of ASAP1 binds directly to Arf1 on lipid nanodiscs. Using interaction restraints and distance constraints from their NMR studies combined with molecular docking and molecular dynamics simulations, the authors generate a structural model for the ASAP1 PH domain in complex with Arf1 on lipid nanodiscs. This integrative structural model reveals that the ASAP1 PH domain binding to Arf1 remodels the switch 1 region of Arf1 to an active conformation. This leads the authors to propose that the PH domain of ASAP1 plays multiple roles in the activation of Arf1 including the recruitment of ASAP1 to the membrane, bringing the GAP domain of ASAP1 in proximity to Arf1 and remodeling the switch 1 region of Arf1. The proposed allosteric role of the PH domain in enhancing the catalytic function of Arf GAPs changes the way we think about the regulation of GTPase proteins. Therefore, this work will be of broad interest to the fields of GTPases, protein allostery and vesicle trafficking.

The manuscript is well written and mostly easy to follow. The data are described well and in sufficient detail. However, some parts of figures are referenced in the text that are not included in the figures. In addition, some controls that more clearly differentiate 1) the contributions of the PH domain in membrane targeting, and 2) potential effects on Arf1 GTP binding, are necessary to address alternative interpretations and would substantially strengthen the conclusions, as outlined in the comments below.

Major comments

1. The authors demonstrate that the PHdZA protein activates Arf1 but at a lower level than the PZA protein. This could be due to differences in the interaction between the PH domains of these two proteins and Arf1 or differences in the abilities of these PH domains to bind membranes. It seems possible that the PHdZA binds membranes more weakly than the PZA protein and this accounts for the reduction in activity seen with the PHdZA protein. Therefore, it would be helpful to test whether PZA and PHdZA are recruited to PIP₂-containing and PC-only LUVs at comparable levels as this would help determine whether differences in the activity of Arf1 arise from variations in membrane recruitment or differences in the activation of Arf1.
2. Figure 1E is missing, although it is referred to in the text in the result section 'To separate the effect of membrane recruitment from other activating mechanisms, we measured...' It seems this reference is intended for the GTPase assay, but the figure itself is not labeled 'E.' In addition, Figure 7 is not correctly referenced in the text. The authors reference Figure 7E and F in the manuscript but Figure 7 only contains panels A-C. Please check the figure labeling and ensure all referenced panels are correctly included and labeled.
3. The conclusion that the PH domain directly stimulates Arf1 based on the 'in trans' experiments in Figure 1D seems somewhat indirect. An alternative interpretation is that the isolated PH and ZA domains may interact with each other in trans, bringing the ZA domain into close proximity with Arf1 and thereby enhancing the GTPase activity of Arf1. One way to directly address this would be to artificially target the ZA domain to membranes using a His-tag and NTA-containing LUVs or replacing the PH domain with a different membrane binding domain such as a PX domain that is predicted to not interact with Arf1. If the GTPase activity of Arf1 is reduced compared to the wild-type PZA construct, this would more directly support the authors' model. On the other hand, if the activity remains comparable to wild-type PZA, it would suggest that the main role of the PH domain may be in recruiting the ZA domain to membranes. The authors do use a different PH domain to recruit the ZA domain to membranes in Figure 1 but it seems possible that other PH domains could activate Arf1 in a similar manner to the ASAP1 PH domain therefore, using a different method to recruit the ZA protein to membranes could significantly strengthen the overall conclusions of the manuscript.
4. Do the authors expect the PH domain of ASAP1 to bind Arf1 in the absence of membranes? Have the authors tried NMR titration experiments similar to Figure 2B in the absence of lipid nanodiscs?
5. The methods largely describe the method to generate the structural model shown in Figure 4 but its not entirely clear if the final structure is the single lowest energy conformation or a bundle of multiple conformations. The authors refer to this as a snapshot representative of the model. If the final structure contains a bundle of different conformations it would be helpful to state how many different conformations and to show an overlay of these different conformations in the supplemental information.
6. The integrative structural model presented by the authors in Figure 4 is quite exciting and would be broadly useful for the field. The authors should make this structural model available in the PDB. The PDB accepts the deposition of integrative structural models.
7. The Results section on page number 12 mentions that Δ N14PH shows near wild-type membrane binding, but the figures

demonstrating this are missing from the manuscript.

8. In Figure 7, the authors demonstrate that mutation of Lys391 in ASAP1 reduces the GTPase activity of Arf1 supporting their proposed model of conformation rearrangement of the Arf1 switch 1 region. However, lysine residues can often aid in membrane binding. Therefore, it is also possible that mutation of Lys391 reduces membrane binding by the ASAP1 PH domain. Have the authors measured the membrane binding of the PH mutant Lys391Ala compared to the wild-type PH domain? This is crucial to conclude that the large reduction in GTPase activity of Arf1 is due to allostery and not reduced membrane binding.

9. Mutations near the nucleotide-binding pocket of Arf1 could also potentially affect GTP binding (e.g., Glu54Ala Arf1). Have the authors measured the level of GTP-bound Arf1 in these mutants using a known Arf1 effector, such as the GGA GAT domain? This would be important to support the claim that the reduced GAP activity is due to allostery alone.

10. While the primary results on the interaction between the PH domain of ASAP1 and Arf1 presented here are valuable, it is unclear if this is a mechanism that might be used by other PH domain containing GAPs. Have the authors performed any sequence alignments to determine whether the Arf1-ASAP1 PH interface identified here is conserved in other Arf family proteins and PH domain-containing Arf-GAPs? This analysis would help identify if similar mechanisms are utilized in significantly broaden the scope and relevance of the manuscript for the vesicular trafficking community. Even if the interface is not conserved in other PH domains this still provides important context for the final conclusions of the paper.

Minor comments

1. The construct design is critical for understanding the paper. I recommend moving this from the supplementary materials to main Figure 1. Additionally, the definition of wtPZA does not mention that it contains the Arf GAP domain; instead, it refers to zinc binding, which may not be immediately clear to non-expert readers. This should be clarified more explicitly.

2. In the Methods section under "GAP activity determined by hydrolysis of [α 32P]GTP bound to Arf on LUVs," includes the sentence: "GAP-induced conversion of myrArf1•GTP to myrArf1•GDP was determined as described previously," but a reference for this is missing.

Reviewer #5

(Remarks to the Author)

Reviewer #6

(Remarks to the Author)

The authors made the discovery that the PH domain of the Arf GAP, ASAP, participates in Arf-GTP substrate binding as well as promoting GTP hydrolysis, in addition to its canonical role as a membrane binding domain. The key data that support this are those where they showed that "in trans" addition of a free-floating PH domain promotes the GTP hydrolysis activity of the free ArfGAP domain, ZA.

The authors then provided structural evidence of the binding interface between the PH domain and Arf. The authors further showed that the PH domain binds tightly to Arf, in contrast with the ZA domain, which binds loosely to the PH-Arf complex. These data supported that the PH domain is the major substrate binding domain, as opposed to the GAP domain.

The authors then went on to show that the PH domain triggers a conformational change in Arf. Firstly, PH "desields" the GTP, potentially rendering hydrolysis more efficient. Secondly, the conformational change is caused by PH destabilizing an internal hydrogen bond network in Arf1.

With the biochemical and structural evidence above, the authors proposed two models to account for the "in trans" experiment and the full PZA experiment in Fig.1, with the major difference in assumption being whether the binding of the catalytic GAP domain ZA following substrate binding with PH is first order or second order. In other words, whether ZA binds to the substrate dependently or independently to the PH already bound to the substrate. The underlying modeling schemes are simple mass-action kinetics of binding between lipids, PH, ZA, and Arf-GTP, as well as Arf-GTP hydrolysis and dissociation of the complex. The model also assumes the difference between 2D search and 3D, which accounts for the effect of PH-facilitated membrane binding, promoting Arf binding. The authors then performed parameter fitting to the experimental data and obtained an optimized parameter set that allowed simulation results to basically fit experiments perfectly. With the modeling results, they show that the mechanism they proposed, where the PH domain facilitates

membrane binding, substrate binding, and prepares the substrate for catalytic activity, can in principle account for the experimental results.

Major comment:

I think this discovery is novel, solid, and worth publishing in Nature Communications. The authors, particularly Paul Randazzo's group, have previously published several articles on noncanonical functions of PH domains, but this is the first time they found that PH enhances enzymatic activity by modifying its substrate. As PH domains are important in a large variety of proteins, this work has a strong impact on the relevant field. In fact, if the title of the manuscript can be more specific to indicate that the allostery happens by PH introducing a conformational change in the substrate, the novelty of the article may be better promoted.

While I am not an expert in structural biology and interpreting NMR results to point out potential caveats in the results, I understand how and why the authors interpret those data and eventually come up with their proposed mechanism. This shows that the writing data presentation is readable enough for a broad audience. The biochemical model they used to summarize the conclusions from their structural data makes perfect sense, and it is not surprising that modeling results would fit well with their biochemical data in Fig.1.

That being said, the authors do need to double-check the cross-referencing between the text and the figures and the figure legends. I pointed out some mis-referencing, but they may not be comprehensive. Please double-check.

Minor comments below:

1. P4, first sentence. Isn't the conclusion from Jian et al 2015 that PHdZA binds better than PZA?
2. P4, first paragraph, please note the full name of PC the first time it appeared.
3. There's no Fig. 1E as referenced at the end of the first results section.
4. In the legend of Fig. 6, there's no panel D. Instead, the NMR data and the cartoon appears to be Panel C left and right.
5. P12 end of second paragraph, there's no Fig. 7EF. I believe they got moved to Fig. SI7E? I suggest putting it back because it better illustrates why the authors picked His405, Phe438, and Glu441 as controls in Fig7, and because it doesn't go well with the rest of FigSI7.
6. I trust the authors that destabilizing the the hydrogen bond network in Arf in Fig7 would cause the conformational change in Fig6. But it is hard for me to visualize it. Would it be possible in Fig7A to superimpose a version of Fig6A that would show the relative position of the hydrogen bond network, the conformational change, and the GTP?
7. Fig. SI8 requires better labeling of which curve is which curve in the figure, and not just in figure legends and SI text.

Version 1:

Reviewer comments:

Reviewer #1

(Remarks to the Author)

The authors have adequately addressed my previous concerns. I recommend publication.

Reviewer #2

(Remarks to the Author)

The authors have satisfactorily addressed my concerns. In my view, the revised manuscript is appropriate for publication.

Reviewer #3

(Remarks to the Author)

Reviewer #4

(Remarks to the Author)

The authors have done a great job addressing all of my concerns.

Reviewer #5

(Remarks to the Author)

Reviewer #6

(Remarks to the Author)

The authors addressed my concerns and modified the text to provide better quality.

REVIEWER COMMENTS

Reviewer #1 (Remarks to the Author):

This manuscript nicely summarizes compelling data from NMR, biochemical, and computational analyses that, together, provide strong evidence for a direct interaction of the Arf1 GTPase with the PH domain of its GAP, ASAP1. The study demonstrates that this interaction results in conformational changes in the switch 1 region of Arf1 to prime GTP hydrolysis. That a GAP PH domain directly activates a GTPase is remarkable because its principal role was thought to be in membrane recruitment and related passive processes. The experiments, simulations, and their analyses are technically sophisticated and state-of-the-art, and the resulting data are internally consistent and, in general, provide a sound basis for the conclusions of the work. However, I have a few concerns that require attention before the manuscript can be accepted for publication.

1) One issue that requires careful consideration is the following: In my reading of the manuscript, the primary 'physical' interaction between ASAP1 and Arf1 on the membrane is through the PH domain, which also recruits the GAP to the membrane, leaving unclear the role of the ArfGAP (Z) domain which in other Ras-related GTPases is known to directly bind at the switch regions of the GTPases. My question then is, will the Z domain of ASAP1 directly engage Arf1 while the PH domain is still bound, or is this some sort of a sequential process? If the former, how is that possible considering that the two GAP domains will have to compete for the same binding region? The proposed model in Fig 8C is confusing in this context, because it is as though the Z domain is not required for "high GAP activity", which is obviously not the case since the PH domain lacks the critical Arg finger.

Response: We have addressed several points that needed further clarification in the manuscript.

Specifically, we have revised Fig. 8C and its legend to precisely indicate the state of Arf-GTP bound to the PH domain as being "primed for GTP hydrolysis" and not "high GAP activity". As the reviewer pointed out, the PH domain lacks the critical Arg finger, originating from the ArfGAP domain. We believe that "primed for GTP hydrolysis" adequately describes Arf* in the Arf:ASAP1 PH complex.

We also clarified in the text that the PH and Arf GAP domains bind distinct, non-overlapping sites, allowing for simultaneous binding. This conclusion is supported by comparing our Arf:PH domain complex structure with the published Arf6:GAP complex (Fig. R1). Further evidence from NMR, MD simulations, and mutagenesis experiments also supports this, with the latter being prepared for a separate manuscript.

Fig. R1: Superimposition of our model for the Arf1: ASAP1 PH complex (blue: Arf; gold: ASAP1 PH) and the published X-ray structure of Arf6 in complex with ASAP3 ZA (pink) showing that the PH and Arf GAP domains bind distinct, non-overlapping sites, allowing for simultaneous binding.

2) A related question involves the ~4-orders of magnitude difference in the GTP hydrolysis rate between the "tandem" and "in trans" systems (Fig 3E). This seems to suggest that the complexes of the tandem and in trans constructs with Arf1 is different and, if that is the case, the mechanism of rate enhancement could also be different. Therefore, how confident can one be about the specific interactions and conformational changes observed by studying the isolated PH domain with Arf1 in explaining the actual mechanism of action of ASAP1 in cells? An attempt is made to explain the different aspects of rate enhancement (membrane recruitment, allostery etc) using high-level mathematical modeling but the structure-mechanistic level explanation remains unconvincing.

Response: We apologize for any prior confusion. Our experiments were designed to thoroughly investigate how the PH domain influences GAP activity, specifically by distinguishing its role in membrane recruitment from other activation mechanisms.

To achieve this, we used different protein constructs with distinct properties:

- wt PZA: This construct features a tandem catalytic ZA domain and its cognate PH domain, enabling it to bind to PIP2-containing membranes and Arf.

- PHdZA: This chimera consists of a catalytic ZA domain linked to the PH domain of PLC δ 1, also allowing binding to PIP2-containing membranes, but not expected to activate Arf.
- ZA: This is the isolated catalytic domain and, crucially, does not bind to membranes and has weak affinity for Arf.

Our initial results, comparing these constructs, suggested a dual role for the PH domain. While membrane recruitment by the PH domain is clearly important—evidenced by significantly higher GAP activity for *wt* PZA (8 orders of magnitude) and PHdZA (4 orders of magnitude) compared to ZA alone. However, given that PHdZA is recruited more efficiently to membranes than is PZA but is less active by 4 orders of magnitude, recruitment cannot explain the total increase in activity observed with the cognate PH domain (as in *wt* PZA). This result led us to examine more direct mechanisms, such as interacting with Arf, to account for the 4 orders of magnitude greater GAP activity of *wt* PZA compared to PHdZA.

To explicitly disentangle the effect of membrane recruitment from these other activating mechanisms, we performed "in trans" GTP hydrolysis assays using isolated PH and ZA domains. In this setup, the PH and ZA domains are not directly linked, meaning membrane recruitment of the catalytic ZA domain is abolished. Strikingly, we observed that simply including the isolated ASAP1 PH domain boosted ZA's activity by 4 orders of magnitude. This "in trans" enhancement demonstrates that allostery—where the PH domain induces a conformational change in Arf to increase catalytic efficiency—is a significant contributor to GAP activity, even without direct membrane recruitment of the catalytic domain. In summary, when the domains are in tandem, both membrane recruitment and allostery contribute to the 8-order-of-magnitude increase in GAP activity. However, in the "in trans" experiment, with membrane recruitment effectively abolished, the 3 to 4-order-of-magnitude increase is solely attributable to allosteric activation.

To further solidify this conclusion, we also performed "in trans" GTP hydrolysis using isolated PLC δ 1 PH and ZA domains during the review process. As anticipated, no rate enhancement was observed in this experiment i.e. PLC δ 1 PH does not bind to Arf in a way that populate a Arf* state, where Arf* is primed for GTP hydrolysis. This result strongly supports the idea that a specific PH domain is required to bind and induce conformational changes in Arf, thus increasing its catalytic efficiency, rather than merely recruiting the GAP to the membrane.

3) Another concern, one that can potentially be addressed by a few control experiments and/simulations, is that all of the NMR, modeling and MD simulations of myr-Arf1 with or without PH were studied in DMPC bilayers. The choice of this lipid is curious because DMPC is not the most representative lipid species in mammalian cells. This requires justification or validation also because membrane biophysical properties including

thickness, lipid packing and electrostatics are likely to modulate the observed intermolecular interactions. Moreover, while the use of simpler model membranes is understandable from a technical point of view, it is important to acknowledge that the actual lipid environment in which Arf1 resides (mostly in endomembranes) is likely to affect many of the current observations, just like the presence of PIP2 alters membrane binding and catalysis.

Response: We considered lipid composition when we set out to do these studies, as well as in previous work in our labs. First, we found kinetic parameters were nearly identical whether using the nanodiscs with the PC/PIP2 mix or large unilamellar vesicles with more complex lipid composition (Li, Soubias et al., *Biochemistry*, 2019). Second, we have examined the effect of lipid composition on the binding of the ASAP1 PH domains to membranes in previous work and found that acyl groups do not contribute significantly, the primary determinant being acid phospholipid content and PIP2 (Roy et al, *J. Biol. Chem*, 2019; Soubias et al, *Sci. Adv*, 2021). We added a sentence in the discussion part to acknowledge the use of model membranes.

4) It is stated in page 14 that "the substrate for the GAP domain is not Arf but Arf bound to the PH domain". This is a strong statement that may require justification beyond the current set of data. For example, are there prior studies supporting this notion, including from GTP hydrolysis experiments with isolated GAPs (more than one GAP) in solution or membrane?

Response: Because we use native Arf, i.e. myristoylated full length, the experiment cannot be done in solution because Arf-GTP is not stable in solution and truncated Arfs are about 1/50,000th efficient as substrates. The idea that Arf bound to a PH domain and, further, to a polypeptide is the substrate for Arf GAPs rather than free Arf is supported by other work. First, which is outside the scope of this paper, we find that PH domains of other Arf GAPs, e.g. AGAPs and ARAPs, also directly contribute to catalysis as we describe for ASAP1. Also, Arf bound to other polypeptides appears to be the substrate for other Arf GAPs, e.g. ArfGAP2 has no activity with free Arf but does have with Arf bound to coatmer, described in (Luo et al., 2009; Luo & Randazzo, 2008) and supported by cryoEM results (Dodonova et al., 2017). We changed the sentence stating that Arf bound to the ASAP1 PH domain is the substrate for the GAP domain, so that it is clear that this idea has only been tested robustly for ASAP1. We speculate in the discussion that it might generalize, briefly mentioning unpublished data but also discussing Arf GAP2.

5) There a few grammatical errors: Examples: "average over of the"; page 8 -- remove 'of', "residues with the", page 10 -- replace "with" by 'within'; Fig 3 E is not labeled.

Response: We thank the reviewer for carefully reading the manuscript. It was corrected in the revised version.

Reviewer #2 (Remarks to the Author):

In this interesting manuscript, Soubias et al. present a structural-mechanistic study of membrane recruitment and allosteric activation by the ASAP1 GAP-PH domain tandem, which mediates membrane recruitment and has GAP activity for Arf1. Quantitative biochemical analyses demonstrate large differences in GAP activity in the presence (and absence) of PIP2-containing liposomes for isolated domains (alone and in combination) vs. the GAP-PH domain tandem. The underlying structural/dynamic basis was thoroughly investigated by NMR in using nanodiscs containing PIP2 and MD simulations, which are used to identify interaction surfaces, construct plausible data-restrained docking models and investigate relevant functional dynamics. Key structural predictions are supported by structure-based mutational analyses. Perhaps the most salient structural-mechanistic insight relates to the observation that whereas most mutations in the PH domain-Arf1 interface have effects that correlate with effects on binding, one notable exception (involving Lys391 in the PH domain of ASAP1 and Glu54 in switch I of Arf1) strongly impairs the catalytic step with little effect on binding, thus supporting a new role for the PH domain in allosteric activation beyond established roles of PH domains in lipid/protein binding and membrane recruitment. Finally, similar relative contributions of membrane recruitment and allosteric activation to GAP activity are delineated using ODE-based mathematical modeling.

Overall, these studies represent an important conceptual advance with respect to the known functional roles of PH domains. The experiments, analyses and modeling appear to be quantitative, rigorous and thorough. The results are convincing and will be of interest to the Arf field as well as the GTPase and PH domain fields more broadly. Indeed, membrane recruitment vs. allosteric activation is relevant to many systems of biological interest but, as far as we are aware, this is the first case for which this important functional distinction is clearly delineated. The manuscript is generally well written, apart from a few instances noted below that would benefit from clarification.

Specific comments:

Does the 'in trans' (PH + ZA) experiment in Figure 1E necessarily separate membrane recruitment from other activation mechanisms? An allosteric activation mechanism requires that the PH domain interact with ZA. Is it also possible that this interaction might in principle enhance membrane recruitment of ZA, or perhaps both? In the absence of additional experiments (e.g., effect of mutations in the PH domain-ZA interface on membrane recruitment vs. allosteric activation) or perhaps ODE-based mathematical modeling, a more nuanced interpretation may be appropriate.

Response: Interesting point. We think our interpretation is OK but maybe our language is imprecise. The PH domain is inducing a conformational change in Arf, priming it for hydrolysis. It is also possible that there is also an effect on the GAP domain but we do not detect any binding of the Arf GAP domain with the PH domain using multiple

approaches in the current experiments (see Fig. 5). In previous work, the association between domains was transient, limited to a small surface on the ankyrin repeats. The exact contacts between the GAP domain and the PH-Arf complex in the catalytic complex are now being examined. As we describe in the manuscript, preliminary studies indicate that the PH domain and Arf GAP domain contact complementary sites on Arf, and therefore, the interpretation that the PH domain induces a conformational change in Arf and that the complex of Arf:PH interacts with the Arf GAP domain through Arf seems the soundest interpretation at this time.

Have the authors considered the alternative possibility that the two patches on the PH domain identified by PREs might arise in the context of a higher order assembly with 2:1 stoichiometry rather than a two-state dynamic equilibrium in a complex with 1:1 stoichiometry? Would such a 2:1 complex (or mixture of 2:1 and 1:1) be consistent or not with the with data and docked models? For example, could both binding sites be occupied simultaneously or is binding mutually exclusive due to steric conflicts? Might be worth discussing briefly.

Response: We thank the reviewer for this interesting comment. We added the model of the β_2/β_3 PH orientation to the SI for reference (Fig. SI4C). As can be seen on the figure below (Fig. R2), the β_5/β_7 and β_2/β_3 PH orientation of the PH domain are mutually exclusive due to significant steric clashes, supporting a two-state dynamic equilibrium model. In addition, the Lys365Asn mutation, which is part of the β_2/β_3 interface did not significantly alter GAP activity (Jian et al, Structure, 2015), contrary to mutations of residues at the β_5/β_7 interface, strongly suggesting that the β_2/β_3 orientation is not functionally relevant.

Fig. R2: Superimposition of Arf in complex with the ASAP1 PH domain in the β_2/β_3 orientation (pink) and in the β_5/β_7 orientation (gold). Steric clashes in the β -sandwich region of the PH domains are apparent.

p. 11, "Notably, switch 1 movement and switch 2 stabilization are key conformational changes that define activation of RAS protein GTPase activity." Seems like a relevant citation(s) would be appropriate here.

Response: We thank the reviewer for carefully reading the manuscript. We added relevant citations to the revised manuscript.

Fig. 8A. Can the authors comment on the modest discrepancy between the fitted ODE models and the data for PZA-PIP2? Is it related mainly to errors for concentration estimation in pM range, details of the mathematical modeling/simultaneous fitting, or something else?

Response: We agree that it might arise from errors in estimating pM concentrations. There are also challenges of global fitting of a finite data set that potentially could lead

to the poor fit of that particular set of data. We also cannot exclude that there are additional contributors to activity, which is an area of active investigation.

Assuming we recall correctly, ring current shifts are highly distance and orientation dependent. Is the ring current shift discussed in the last paragraph on p. 11 consistent with the structural model(s)?

Response: Thank you for pointing that out. The reviewer is correct. Unfortunately we can't be more precise with our interpretation because of the transient nature of the interactions.

Minor:

P. 5, "which both binds" should be "which both bind"

Fig. 1E, the label "E" is missing

Define acronyms on first usage (e.g., GEF, PC, etc)

There are a few other typos and inconsistencies throughout the manuscript. For example, some headings end with a period, other don't.

Response: We thank the reviewer for carefully reading the manuscript. It was corrected in the revised version of the manuscript.

Reviewer #3 (Remarks to the Author):

Reviewer #4 (Remarks to the Author):

Arf1 GTPase regulates vesicular trafficking in cells. Arf1 has no GTPase activity on its own and instead requires binding to GTPase activating proteins (GAPs) to allow it to hydrolyze GTP to GDP. It was previously thought that GAP domains would be sufficient for the activation of Arf1. However, recent studies have demonstrated that the GAP domain of ASAP1 by itself is not sufficient to activate Arf1. This suggested that other domains in ASAP1 could play important roles in activating Arf1. However, it was not clear how these domains would help activate Arf1. The manuscript by Soubias et al.

identifies that the membrane binding PH domain of ASAP1 is required for the activation of Arf1. Using NMR studies, the authors demonstrate that the PH domain of ASAP1 binds directly to Arf1 on lipid nanodiscs. Using interaction restraints and distance constraints from their NMR studies combined with molecular docking and molecular dynamics simulations, the authors generate a structural model for the ASAP1 PH domain in complex with Arf1 on lipid nanodiscs. This integrative structural model reveals that the ASAP1 PH domain binding to Arf1 remodels the switch 1 region of Arf1 to an active conformation. This leads the authors to propose that the PH domain of ASAP1 plays multiple roles in the activation of Arf1 including the recruitment of ASAP1 to the membrane, bringing the GAP domain of ASAP1 in proximity to Arf1 and remodeling the switch 1 region of Arf1. The proposed allosteric role of the PH domain in enhancing the catalytic function of Arf GAPs changes the way we think about the regulation of GTPase proteins. Therefore, this work will be of broad interest to the fields of GTPases, protein allostery and vesicle trafficking.

The manuscript is well written and mostly easy to follow. The data are described well and in sufficient detail. However, some parts of figures are referenced in the text that are not included in the figures. In addition, some controls that more clearly differentiate 1) the contributions of the PH domain in membrane targeting, and 2) potential effects on Arf1 GTP binding, are necessary to address alternative interpretations and would substantially strengthen the conclusions, as outlined in the comments below.

Major comments

1. The authors demonstrate that the PHdZA protein activates Arf1 but at a lower level than the PZA protein. This could be due to differences in the interaction between the PH domains of these two proteins and Arf1 or differences in the abilities of these PH domains to bind membranes. It seems possible that the PHdZA binds membranes more weakly than the PZA protein and this accounts for the reduction in activity seen with the PHdZA protein. Therefore, it would be helpful to test whether PZA and PHdZA are recruited to PIP₂-containing and PC-only LUVs at comparable levels as this would help determine whether differences in the activity of Arf1 arise from variations in membrane recruitment or differences in the activation of Arf1.

Response: Thank you for your comment. We concur with the reviewer that reduced membrane binding by PHdZA could potentially lead to weaker GAP activity. However, this possibility was addressed in our earlier study (Jian et al., Structure, 2015), where we demonstrated that PHdZA binds to PIP₂-containing LUVs with 5–10 times greater affinity than PZA. This rules out weaker membrane association as the cause of the reduced activity observed with PHdZA.

Furthermore, during the course of this review, we conducted "in trans" GTP hydrolysis assays using isolated PLCδ1 PH and ASAP1 ZA domains (Fig. S12A). These experiments showed no enhancement in hydrolysis rates, confirming that the PHdZA chimera functions solely as a reporter of membrane recruitment.

2. Figure 1E is missing, although it is referred to in the text in the result section 'To separate the effect of membrane recruitment from other activating mechanisms, we measured...' It seems this reference is intended for the GTPase assay, but the figure itself is not labeled 'E.' In addition, Figure 7 is not correctly referenced in the text. The authors reference Figure 7E and F in the manuscript but Figure 7 only contains panels A-C. Please check the figure labeling and ensure all referenced panels are correctly included and labeled.

Response: We thank the reviewer for carefully reading the manuscript. It was corrected in the revised version of the manuscript.

3. The conclusion that the PH domain directly stimulates Arf1 based on the 'in trans' experiments in Figure 1D seems somewhat indirect. An alternative interpretation is that the isolated PH and ZA domains may interact with each other in trans, bringing the ZA domain into close proximity with Arf1 and thereby enhancing the GTPase activity of Arf1. One way to directly address this would be to artificially target the ZA domain to membranes using a His-tag and NTA-containing LUVs or replacing the PH domain with a different membrane binding domain such as a PX domain that is predicted to not interact with Arf1. If the GTPase activity of Arf1 is reduced compared to the wild-type PZA construct, this would more directly support the authors' model. On the other hand, if the activity remains comparable to wild-type PZA, it would suggest that the main role of the PH domain may be in recruiting the ZA domain to membranes. The authors do use a different PH domain to recruit the ZA domain to membranes in Figure 1 but it seems possible that other PH domains could activate Arf1 in a similar manner to the ASAP1 PH domain therefore, using a different method to recruit the ZA protein to membranes could significantly strengthen the overall conclusions of the manuscript.

Response: We thank the reviewer for the insightful comment. As suggested, we assessed Arf1 GTPase activity in a system comprising His-tagged ZA, Arf1, and vesicles containing PIP2 along with headgroup-modified Ni-NTA lipids. As shown in Fig. R2, GTPase activity is markedly reduced compared to the wild-type PZA, which supports our interpretation.

As the reviewer noted, we employ a different PH domain to recruit the ZA domain—specifically, the PHdZA chimera construct. In line with the reviewer's suggestion, it is indeed plausible that other PH domains might activate Arf1 in a manner similar to the ASAP1 PH domain. To explore this possibility, we conducted "in trans" GTP hydrolysis assays using isolated PLC δ 1 PH and ZA domains during the review process. These experiments revealed no enhancement in hydrolysis rates, thereby ruling out the ability of the PLC δ 1 PH domain to activate Arf1 (Fig. SI2A).

Fig. R2. GAP activity measured for His-tagged ZA and ZA and Ni-NTA lipids containing LUV (5 mol % PIP2, 5 mol % 18:1 DGS-NTA(Ni) lipids, all other conditions are the same). For comparison, GAP activity curves measured for PZA and PdZA measured in PIP2 containing LUV (5 mol% PIP2) are shown.

4. Do the authors expect the PH domain of ASAP1 to bind Arf1 in the absence of membranes? Have the authors tried NMR titration experiments similar to Figure 2B in the absence of lipid nanodiscs?

Response: We thank the reviewer for the suggestions. However, the suggested experiment is not possible as native Arf-GTP is not stable in solution, Arf-GDP doesn't detectably bind to the PH domain and truncated Arfs, which are stable in solution when GTP bound, have about 1/50,000th the activity with the GAP and still require a membrane to observe the little activity that is detected.

5. The methods largely describe the method to generate the structural model shown in Figure 4 but its not entirely clear if the final structure is the single lowest energy conformation or a bundle of multiple conformations. The authors refer to this as a snapshot representative of the model. If the final structure contains a bundle of different conformations it would be helpful to state how many different conformations and to show an overlay of these different conformations in the supplemental information.

Response: We thank the reviewer for the thoughtful remark. The structure shown in Fig. 4A represents a snapshot of the membrane-associated complex model derived from our MD simulations. As detailed in the text, we used the full ensemble of MD conformations to cross-validate the experimentally obtained PREs (or vice-versa).

Given the strong agreement between the experimental PRE data and those back calculated from the MD trajectories, we did not apply further filtering of the MD results based on the experimental data. To illustrate this, we have included a supplemental figure (Fig. SI5) depicting a bundle of representative structures.

6. The integrative structural model presented by the authors in Figure 4 is quite exciting and would be broadly useful for the field. The authors should make this structural model available in the PDB. The PDB accepts the deposition of integrative structural models.

Response: Thank you. The deposition of the integrative model in the PDB has been initiated and the MD trajectories will be made available upon request.

7. The Results section on page number 12 mentions that $\Delta N14PH$ shows near wild-type membrane binding, but the figures demonstrating this are missing from the manuscript.

Response: Wild type and $\Delta N14PH$ have indeed almost equal binding affinity to PIP2-containing membranes. We previously published the result in Roy et al., (Fig. 7E) which we refer to.

8. In Figure 7, the authors demonstrate that mutation of Lys391 in ASAP1 reduces the GTPase activity of Arf1 supporting their proposed model of conformation rearrangement of the Arf1 switch 1 region. However, lysine residues can often aid in membrane binding. Therefore, it is also possible that mutation of Lys391 reduces membrane binding by the ASAP1 PH domain. Have the authors measured the membrane binding of the PH mutant Lys391Ala compared to the wild-type PH domain? This is crucial to conclude that the large reduction in GTPase activity of Arf1 is due to allostery and not reduced membrane binding.

Response: Thank you for your comment. We assessed the membrane binding of the PZA Lys391Ala mutant in comparison to wild-type PZA using a sucrose-loaded vesicle centrifugation assay performed under the same experimental conditions as the GAP activity assay. As shown, replacing Lys391 with alanine does not alter membrane association. Combined with the observation that this substitution has only a minimal effect on the binding of the mutant PH domain to Arf, we believe this supports our interpretation and underscores the critical role of Lys391 in the allosteric mechanism. This clarification has been included in the revised manuscript (Fig. SI8E).

9. Mutations near the nucleotide-binding pocket of Arf1 could also potentially affect GTP binding (e.g., Glu54Ala Arf1). Have the authors measured the level of GTP-bound Arf1 in these mutants using a known Arf1 effector, such as the GGA GAT domain? This would be important to support the claim that the reduced GAP activity is due to allostery alone.

Response: Thank you for your comment. Rather than the GGA pull down assay, which is not quantitative, we assessed the structural integrity and concentrations of the recombinant mutant Arf proteins by titrating nucleotide binding sites, measured by GTP binding in a filter trapping assay using ^{35}S GTP γ S as a tracer which we now state in methods. The assay is described in multiple publications, e.g. (Randazzo et al., 1995). Each mutant was assayed by nucleotide binding before all GAP assays to ensure that experiments were performed at the same concentration of active proteins.

10. While the primary results on the interaction between the PH domain of ASAP1 and Arf1 presented here are valuable, its unclear if this is a mechanism that might be used by other PH domain containing GAPs. Have the authors performed any sequence alignments to determine whether the Arf1-ASAP1 PH interface identified here is conserved in other Arf family proteins and PH domain-containing Arf-GAPs? This analysis would help identify if similar mechanisms are utilized in significantly broaden the scope and relevance of the manuscript for the vesicular trafficking community. Even if the interface is not conserved in other PH domains this still provides important context for the final conclusions of the paper.

Response: We appreciate the reviewer's forward-looking perspective. We are equally enthusiastic about investigating whether the mechanisms identified in this study extend to other PH domain-containing proteins. In Figure 8 of the original manuscript, we addressed this by aligning the structure of the ASAP1 PH domain with structural models of three additional ArfGAP PH domains—specifically those of ACAP1, AGAP1, and ARAP1. This comparison revealed that the QVK motif, which includes Lys391 in ASAP1, is conserved in these domains, suggesting a potentially shared mechanism. This hypothesis is currently being explored through ongoing studies in our lab, but are outside the scope of this manuscript.

Minor comments

1. The construct design is critical for understanding the paper. I recommend moving this from the supplementary materials to main Figure 1. Additionally, the definition of wtPZA does not mention that it contains the Arf GAP domain; instead, it refers to zinc binding, which may not be immediately clear to non-expert readers. This should be clarified more explicitly.

Response: Fig. 1. has been modified to add the construct design.

2. In the Methods section under "GAP activity determined by hydrolysis of [α 32P]GTP bound to Arf on LUVs," includes the sentence: "GAP-induced conversion of myrArf1•GTP to myrArf1•GDP was determined as described previously," but a reference for this is missing.

Response: Thank you. The reference was added.

Reviewer #5 (Remarks to the Author):

Reviewer #6 (Remarks to the Author):

The authors made the discovery that the PH domain of the Arf GAP, ASAP, participates in Arf-GTP substrate binding as well as promoting GTP hydrolysis, in addition to its canonical role as a membrane binding domain. The key data that support this are those where they showed that “in trans” addition of a free-floating PH domain promotes the GTP hydrolysis activity of the free ArfGAP domain, ZA.

The authors then provided structural evidence of the binding interface between the PH domain and Arf. The authors further showed that the PH domain binds tightly to Arf, in contrast with the ZA domain, which binds loosely to the PH-Arf complex. These data supported that the PH domain is the major substrate binding domain, as opposed to the GAP domain.

The authors then went on to show that the PH domain triggers a conformational change in Arf. Firstly, PH “deshields” the GTP, potentially rendering hydrolysis more efficient. Secondly, the conformational change is caused by PH destabilizing an internal hydrogen bond network in Arf1.

With the biochemical and structural evidence above, the authors proposed two models to account for the “in trans” experiment and the full PZA experiment in Fig.1, with the major difference in assumption being whether the binding of the catalytic GAP domain ZA following substrate binding with PH is first order or second order. In other words, whether ZA binds to the substrate dependently or independently to the PH already bound to the substrate. The underlying modeling schemes are simple mass-action kinetics of binding between lipids, PH, ZA, and Arf-GTP, as well as Arf-GTP hydrolysis and dissociation of the complex. The model also assumes the difference between 2D search and 3D, which accounts for the effect of PH-facilitated membrane binding, promoting Arf binding. The authors then performed parameter fitting to the experimental data and obtained an optimized parameter set that allowed simulation results to basically fit experiments perfectly. With the modeling results, they show that the mechanism they proposed, where the PH domain facilitates membrane binding,

substrate binding, and prepares the substrate for catalytic activity, can in principle account for the experimental results.

Major comment:

I think this discovery is novel, solid, and worth publishing in Nature Communications. The authors, particularly Paul Randazzo's group, have previously published several articles on noncanonical functions of PH domains, but this is the first time they found that PH enhances enzymatic activity by modifying its substrate. As PH domains are important in a large variety of proteins, this work has a strong impact on the relevant field. In fact, if the title of the manuscript can be more specific to indicate that the allostery happens by PH introducing a conformational change in the substrate, the novelty of the article may be better promoted.

While I am not an expert in structural biology and interpreting NMR results to point out potential caveats in the results, I understand how and why the authors interpret those data and eventually come up with their proposed mechanism. This shows that the writing data presentation is readable enough for a broad audience. The biochemical model they used to summarize the conclusions from their structural data makes perfect sense, and it is not surprising that modeling results would fit well with their biochemical data in Fig.1.

That being said, the authors do need to double-check the cross-referencing between the text and the figures and the figure legends. I pointed out some mis-referencing, but they may not be comprehensive. Please double-check.

Minor comments below:

1. P4, first sentence. Isn't the conclusion from Jian et al 2015 that PHdZA binds better than PZA?

Response: The reviewer is correct. We modified the text.

2. P4, first paragraph, please note the full name of PC the first time it appeared.

Response: Thank you for thoroughly reading the manuscript. It was corrected in the revised manuscript.

3. There's no Fig. 1E as referenced at the end of the first results section.

Response: This was corrected.

4. In the legend of Fig. 6, there's no panel D. Instead, the NMR data and the cartoon appears to be Panel C left and right.

Response: This was corrected.

5. P12 end of second paragraph, there's no Fig. 7EF. I believe they got moved to Fig. SI7E? I suggest putting it back because it better illustrates why the authors picked His405, Phe438, and Glu441 as controls in Fig7, and because it doesn't go well with the rest of FigSI7.

Response: Due to lack of space, we could not incorporate what was Fig. SI7E to Fig. 7. For clarity, Fig SI7E is now Fig. SI9 and the text has been modified as follows: "In contrast, mutations at interface residues His405 and Phe438 showed activity reductions (5- and 15-fold) more proportional to their expected binding effects. Glu441 mutations, predicted to be outside the interface, had no effect (Fig. 7C, Fig. SI9, Table SI4, 5)."

6. I trust the authors that destabilizing the the hydrogen bond network in Arf in Fig7 would cause the conformational change in Fig6. But it is hard for me to visualize it. Would it be possible in Fig7A to superimpose a version of Fig6A that would show the relative position of the hydrogen bond network, the conformational change, and the GTP?

Response: Fig 7A has been updated to help visualization.

7. Fig. SI8 requires better labeling of which curve is which curve in the figure, and not just in figure legends and SI text.

Response: Thank you for your comment. This was modified.

- Dodonova, S. O., Aderhold, P., Kopp, J., Ganeva, I., Röhling, S., Hagen, W. J. H., Sinning, I., Wieland, F., & Briggs, J. A. G. (2017). 9Å structure of the COPI coat reveals that the Arf1 GTPase occupies two contrasting molecular environments. *Elife*, 6, e26691. <https://doi.org/10.7554/eLife.26691>
- Luo, R., Ha, V. L., Hayashi, R., & Randazzo, P. A. (2009). Arf GAP2 is positively regulated by coatamer and cargo. *Cell Signal*, 21(7), 1169-1179. <https://doi.org/10.1016/j.cellsig.2009.03.006>
- Luo, R., & Randazzo, P. A. (2008). Kinetic analysis of Arf GAP1 indicates a regulatory role for coatamer. *J Biol Chem*, 283(32), 21965-21977. <https://doi.org/10.1074/jbc.M802268200>
- Randazzo, P. A., Terui, T., Sturch, S., Fales, H. M., Ferrige, A. G., & Kahn, R. A. (1995). The myristoylated amino terminus of ADP-ribosylation factor 1 is a phospholipid- and GTP-sensitive switch. *J Biol Chem*, 270(24), 14809-14815.